# Reversions mask the contribution of adaptive evolution in microbiomes

**Paul A Torrillo[1,2], Tami D Lieberman[1,2,3,4]\***

[1]Institute for Medical Engineering and Sciences, Massachusetts Institute of Technology, Cambridge, United States; [2]Department of Civil and Environmental Engineering, Massachusetts Institute of Technology, Cambridge, United States; [3]Broad Institute of MIT and Harvard, Cambridge, United States; [4]Ragon Institute of MGH, MIT and Harvard, Cambridge, United States

**Abstract** When examining bacterial genomes for evidence of past selection, the results depend heavily on the mutational distance between chosen genomes. Even within a bacterial species, genomes separated by larger mutational distances exhibit stronger evidence of purifying selection as assessed by $d_N/d_S$, the normalized ratio of nonsynonymous to synonymous mutations. Here, we show that the classical interpretation of this scale dependence, weak purifying selection, leads to problematic mutation accumulation when applied to available gut microbiome data. We propose an alternative, adaptive reversion model with opposite implications for dynamical intuition and applications of $d_N/d_S$. Reversions that occur and sweep within-host populations are nearly guaranteed in microbiomes due to large population sizes, short generation times, and variable environments. Using analytical and simulation approaches, we show that adaptive reversion can explain the $d_N/d_S$ decay given only dozens of locally fluctuating selective pressures, which is realistic in the context of *Bacteroides* genomes. The success of the adaptive reversion model argues for interpreting low values of $d_N/d_S$ obtained from long timescales with caution as they may emerge even when adaptive sweeps are frequent. Our work thus inverts the interpretation of an old observation in bacterial evolution, illustrates the potential of mutational reversions to shape genomic landscapes over time, and highlights the importance of studying bacterial genomic evolution on short timescales.

**\*For correspondence:**
tami@mit.edu

## Editor's evaluation

This valuable study addresses the interpretation of patterns of synonymous and nonsynonymous diversity in microbial genomes. The authors present solid theoretical and computational evidence that adaptive mutations that revert the amino acids to an earlier state can significantly impact the observed ratios of synonymous and nonsynonymous mutations in human commensal bacteria. This article will be of interest to microbiologists with a background in evolution and to researchers studying the human microbiome.

## Introduction

Understanding evolutionary pressures acting upon bacterial populations is crucial for predicting the emergence and future virulence of pathogens (*Culyba and Van Tyne, 2021*), modeling strategies to combat antimicrobial resistance (*Davies and Davies, 2010*), and designing genetically modified organisms (*Castle et al., 2021*). Bacteria can adapt at rapid rates due to their short generation times and large population sizes. Indeed, the rapid evolutionary potential of the microbiome has been proposed to assist in the dietary transitions of mammals (*Kolodny and Schulenburg, 2020*). However, the vast majority of possible mutations do not increase bacterial fitness and instead result in a neutral

or deleterious effect (*Kimura, 1977*; *Davies et al., 1999*; *Jolley et al., 2000*; *Dingle et al., 2001*). Metrics that estimate the directionality and intensity of past selection at genomic loci of interest have thus become critical tools in modern microbiology and biology more generally.

The normalized ratio of nonsynonymous (N) to synonymous (S) substitutions, known as $d_N/d_S$ or the $K_A/K_S$ ratio, is a widely used indicator of past selection (*Jukes and Cantor, 1969*; *Kryazhimskiy and Plotkin, 2008*; *Barber and Elde, 2014*). Nonsynonymous substitutions change the encoded amino acid and thus are considered likely to impact a protein's function, while synonymous substitutions do not affect the encoded amino acid and are therefore considered effectively neutral, with limited exceptions (*Nowick et al., 2019*). To account for the fact that nonsynonymous mutations are more likely than synonymous mutations based on the genomic code (~3× on average; *Yang and Nielsen, 2000*), the values '$d_N$' and '$d_S$' normalize mutation counts to available sites on the genome. The $d_N/d_S$ ratio therefore summarizes past selection on a genetic sequence, which could be a whole genome, pathway, gene, functional domain, or nucleotide; notably values of $d_N/d_S$ averaged genome-wide can obscure signatures of adaptive evolution on other portions of the genome (*Loo et al., 2020*; *Ho et al., 2005*; *Peterson and Masel, 2009*). A $d_N/d_S$ ratio of >1 indicates the dominance of past adaptive evolution (i.e., directional selection) while a ratio of <1 traditionally implies past selection against amino acid change (purifying selection).

Early sequencing work comparing bacterial genomes of the same species reported relatively low $d_N/d_S$ values across the whole genome (<0.15) (*Jolley et al., 2000*; *Dingle et al., 2001*). These observations, obtained from comparing distant bacteria within each species, indicated a strong predominance of purifying selection. However, as it became economically feasible to sequence organisms separated by fewer mutations and therefore less evolutionary time, a contrasting pattern emerged in which high $d_N/d_S$ values (~1) were found between closely related strains (*Feil et al., 2003*; *Baker et al., 2004*). Recent work in the human microbiome has confirmed such results and furthered the contrast between timescales by finding values of $d_N/d_S > 1$ (*Garud et al., 2019*; *Lieberman et al., 2011*; *Shoemaker et al., 2022*). The timescale dependence of $d_N/d_S$ has been mainly attributed to the ongoing action of purifying selection (*Garud et al., 2019*), a model first proposed by *Rocha et al., 2006*. According to this model, weak purifying selection (or locally inactive purifying selection; *Loo et al., 2020*) allows for an initially inflated $d_N/d_S$ ratio as deleterious mutations that will eventually be purged remain in the population. As time progresses and purifying selection continuously operates, the $d_N/d_S$ ratio decreases (*Loo et al., 2020*; *Garud et al., 2019*; *Rocha et al., 2006*). However, multiple studies have observed genome-wide values of $d_N/d_S > 1$ in these same microbial systems, with values substantially >1 in key genes, which are simply unaccounted for in the purifying model (*Garud et al., 2019*; *Lieberman et al., 2011*; *Marvig et al., 2015*; *Zhao et al., 2019*; *Zhao et al., 2020*).

Here, we demonstrate fundamental flaws in the purifying selection model in the context of the large within-person population sizes typical to the human microbiome and many bacterial infections (>$10^{12}$ bacteria/person). We use analytical, simulation-based, and genomic approaches to support a contrasting model for the timescale dependence of $d_N/d_S$, in which adaptive evolution predominates but is not apparent on long-timescales due to adaptive reversion. The comparative success of the reversion model suggests that the study of closely related bacteria is needed to fully understand evolutionary dynamics.

## Results
### A model of purifying selection that fits the data reveals unrealistic parameters

Explaining the timescale dependence of $d_N/d_S$ through an exclusively purifying selection model poses several challenges. Firstly, fitting observed data with purifying selection requires a preponderance of mutations with extraordinarily small effects on fitness (selective coefficients, *s*), which are challenging to eliminate effectively (*Haigh, 1978*). Secondly, the occurrence of an adaptive event during the extensive time required to purge weakly deleterious mutations interrupts the purification of such mutations. Lastly, neutral bottlenecking processes, such as those observed during host-to-host transmission, exacerbate the accumulation of deleterious mutations. For most of this section, we will disregard these last two complications and focus on the problem of small *s*. To provide clarity, we first detail the classic purifying selection model.

Mutations can be divided into three classes, the first two of which accumulate at a constant rate per unit of time: synonymous mutations ($S$), neutral nonsynonymous mutations ($N_{neut}$), and non-neutral, transient, nonsynonymous mutations ($N_{transient}$). We restate the timescale dependence of $d_N/d_S$ as the observation that, in a population starting from a single wild type (WT) cell, the average number of non-neutral nonsynonymous mutations per cell in the population ($\overline{N}_{transient}$) increases and then asymptotes. Assuming an infinitely large population size and an infinite genome size (to circumvent saturation of mutations), the exclusive purifying selection model (*Garud et al., 2019*; *Rocha et al., 2006*) can thus be written as

$$\frac{d_N}{d_S} = \frac{\overline{N}_{neut} + \overline{N}_{transient}}{3\overline{S}} \tag{1}$$

and

$$d\overline{N}_{transient} = U_N dt - s\overline{N}_{transient}dt. \tag{2}$$

Here, $U_N$ is the non-neutral mutation rate per core genome per generation, $s$ is the selective disadvantage of a non-neutral nonsynonymous mutation (or the harmonic mean of such mutations; see Appendix 1, Section 1.1), and $t$ is the number of generations. The 3 in the denominator of *Equation 1* accounts for the discrepancy in the number of potential nonsynonymous and synonymous sites (*Yang and Nielsen, 2000*). We solve for $\overline{N}_{transient}$ by assuming $\overline{N}_{transient}\left(t = 0\right) = 0$ to obtain:

$$\overline{N}_{transient}\left(t\right) = \frac{U_N\left(1 - e^{-st}\right)}{s}. \tag{3}$$

We further simplify and combine these equations to create an equation for $d_N/d_S$ with only two parameters as previously done (*Garud et al., 2019*). First, since $d_N/d_S$ plateaus with time (*Figure 1a*), we have $\lim\limits_{t\to\infty} \frac{\overline{N}_{neut} + \overline{N}_{transient}}{3\overline{S}} = \frac{\overline{N}_{neut}}{3\overline{S}} = \alpha$. Conveniently, $\alpha$ represents both the asymptote of $d_N/d_S$ and the proportion of nonsynonymous mutations that are neutral. This allows us to leave only $s$ as the other free parameter, obtaining (see Appendix 1, Section 1.1)

$$\frac{d_N}{d_S} = \alpha + \left(1 - \alpha\right)\frac{1 - e^{-st}}{st}. \tag{4}$$

As sequence analysis is not privy to the actual number of generations, we approximate $t$ assuming that synonymous mutations accumulate according to a molecular clock ($t = \frac{d_S}{2\left(1/4\right)\mu}$), where μ is the mutation rate per generation per base pair, ¼ represents the proportion of random mutations that are synonymous (*Yang and Nielsen, 2000*), and 2 accounts for the fact that divergence is a measure between a pair of genomes. As selection and mutation are both in units per time, any change in μ results in a corresponding change in $s$. Both model fits and consequences are largely dependent on the ratio of these two variables (more on this below), and thus are not sensitive to the choice of μ. We use a relatively high mutation rate of $10^{-9}$ per base pair per generation (*Drake, 1991*; *Barrick and Lenski, 2013*) as lower rate would imply even weaker purifying selection.

Fitting the data from *Garud et al., 2019*, we infer median values of α ≈ 0.10 (0.09–0.14) and s ≈ 3.5 × $10^{-5}$ (2.6 × $10^{-5}$-6.5 × $10^{-5}$) across all species ('Methods', *Figure 1a*). Aggregating all of the data at once results in a similar optimal fit of α ≈ 0.11 and s ≈ 2.8 × $10^{-5}$. The similarity across the 10 species is perhaps not surprising, given that all are human gut residents of the order *Bacteroidales*; these values are also in line with the values obtained previously from aggregating across all species (*Garud et al., 2019*). These values indicate a model in which only ~10% of nonsynonymous mutations are neutral and the remaining ~90% are so weakly deleterious that they are beyond the limit of detection of any experimental method to date ($s \gtrsim 10^{-3}$) (*Gallet et al., 2012*). Higher values of $s$ that better reflect experimental observations (*Kibota and Lynch, 1996*; *Trindade et al., 2010*; *Robert et al., 2018*) result in poor fits to the data (*Figure 1—figure supplement 3*). While the implied proportion of deleterious mutations may seem high, deep mutational scanning experiments have revealed that most amino acid-changing mutations in essential genes are deleterious enough to be measured in the lab (*Kelsic et al., 2016*; *Dewachter et al., 2023*); complex real-world environments are expected to constrain an even larger fraction of the genome.

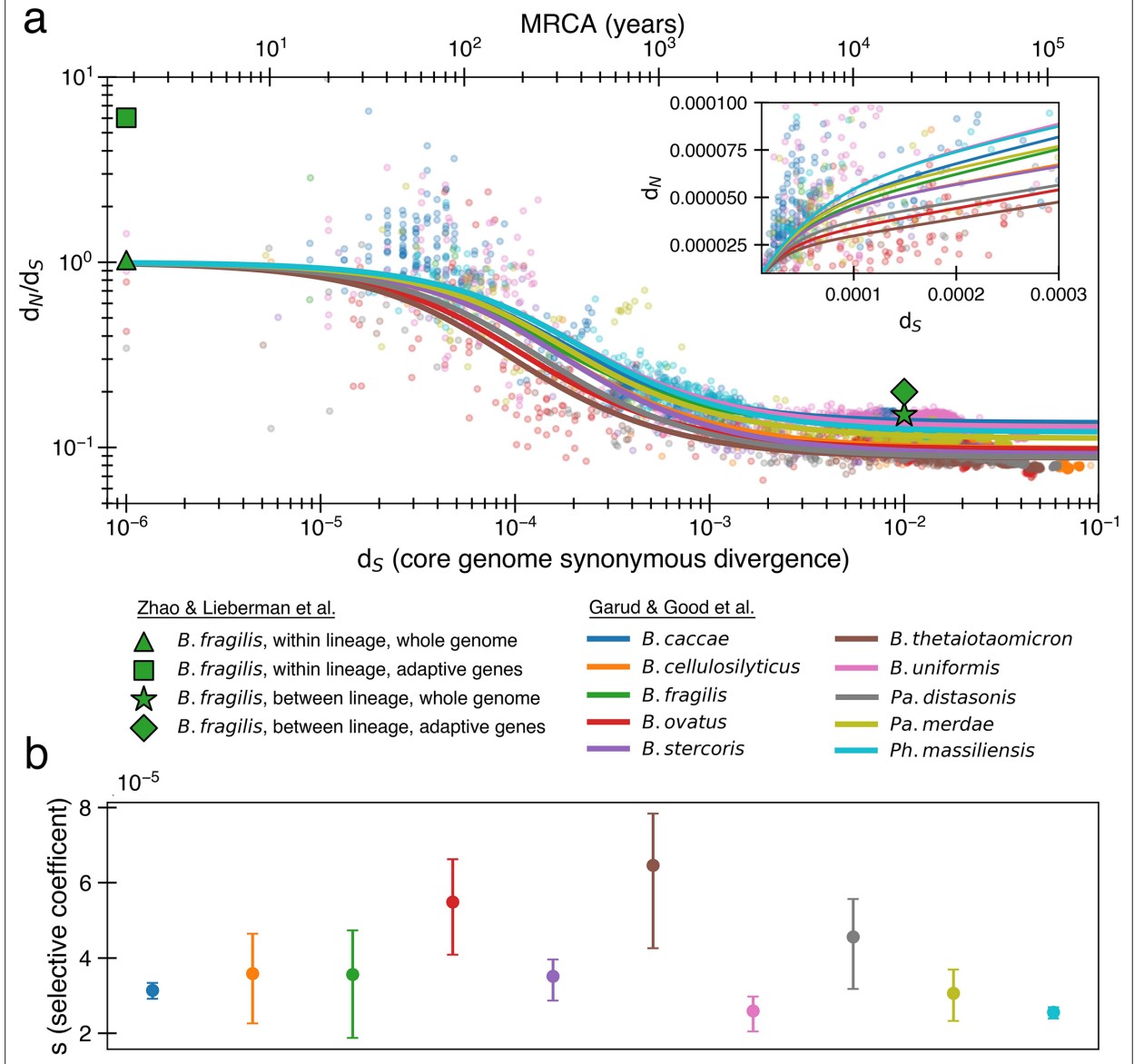

**Figure 1.** The previously proposed explanation for the time dependence of $d_N/d_S$ is weak purifying selection. (**a**) Time signature of $d_N/d_S$ as depicted by data points derived from the studies by *Garud et al., 2019* and *Zhao et al., 2019*. Each dot represents a pairwise comparison between the consensus sequence from two gut microbiomes as computed by *Garud et al., 2019*, using only the top 10 species based on the quality of data points (see 'Methods'). Where the high initial value of $d_N/d_S$ begins to become the low asymptotic value of $d_N/d_S$ occurs at approximately $d_S = \frac{\mu_S}{s}$. Fit lines were derived from these points using *Equation 4* to depict the trend. The median $R^2$ is 0.81 (range 0.54–0.94). Corresponding data from *Zhao et al., 2019* confirms these observed trends, demonstrating high levels of $d_N/d_S$ at short timescales and low levels at longer timescales. Adaptive genes are from *Zhao et al., 2019* and are defined as those that have high $d_N/d_S$ values in multiple lineages. Insets: $d_N$ vs. $d_S$ on a linear scale. Note that the data was fit to minimize variance in the logarithmic scale, not the linear scale, so the fit is not expected to be as good for the inset. See *Figure 1—figure supplement 1* for minimizing variance on a linear scale. See *Figure 1—figure supplement 2* for all species on separate panels. (**b**) Values of *s* from the output of 999 standard bootstrap iterations of curve fitting, conducted with replacement, demonstrate that only small values of the average selective coefficient can fit the data.

The online version of this article includes the following figure supplement(s) for figure 1:

**Figure supplement 1.** The purifying selection model predicts even weaker purifying selection when fitting to nonlogarithmic $d_N/d_S$.

**Figure supplement 2.** Purifying selection model fits by individual species.

**Figure supplement 3.** Larger selective coefficients rapidly lose explanatory power.

In finite populations, the presence of so many weakly deleterious mutations becomes quickly problematic. When $s$ is smaller than $U_N$, organisms without any deleterious mutations (or with the fewest number of deleterious mutations, the 'least-loaded class'; *Haigh, 1978*) can be easily lost from a finite population before they outcompete less fit organisms and fitness decay begins to occur. The likelihood of loss depends on the population size and mutation-selection balance ($U_N/s$), a parameter that estimates the average number of deleterious mutations per cell relative to the least-loaded class. Given a core genome of $L = 1.5 \times 10^6$ bp that can acquire deleterious mutations, we then expect 0.001 new deleterious mutations per genome per generation ($U_N = \frac{3}{4}(1 - \alpha)L\mu$). Thus, the value of $U_N/s$ for the above fits is ~29, indicating that most cells in the population contain dozens of deleterious mutations (see Appendix 1, Section 1.2). With this value of the mutation-selection balance parameter, the frequency of mutation-free organisms in a population is extremely small, even for a population that starts without any deleterious mutations ($<10^{-12}$ after 100,000 generations). If the flexible genome also contains deleterious mutations, the least-loaded class is pushed down even further. Simulations substantiate this prediction of mutation accumulation and decrease in frequency of the wild type (*Figure 2a*, 'Methods').

The time until the least-loaded class is completely lost from the population depends on the strength of genetic drift. The strength of genetic drift is inversely proportional to population size in well-mixed populations (*Gillespie, 2004*), and in less well-mixed or otherwise nonideal populations, is inversely proportional to a smaller parameter, the effective population size, $N_e$. $N_e$ is often estimated by assessing polymorphisms in a population (*Gillespie, 2004*) but is hard to estimate from data following a recent bottleneck. Because each individual's gut microbiome is thought to be well mixed (census size = $10^{13}$) (*Sender et al., 2016*), it has been recently argued that $N_e \approx 10^{11}$ reflects drift processes for dominant gut species (*Ghosh and Good, 2022*; *Labavić et al., 2022*). On the other hand, lower values of $N_e \approx 10^9$ or less have been estimated for global populations of bacteria (*Bobay and Ochman, 2018*) because of the slow rates of bacterial transmission across people. While this decrease in $N_e$ when increasing scales may seem paradoxical, we note this use of $N_e$ only reflects the magnitude of the force of drift; for other calculations in nonideal populations, census population size or other parameters should be used.

Without extremely large values of $N_e$, the least-loaded class will be lost recurrently, rapidly lowering the fitness of the population (i.e., Muller's ratchet; *Haigh, 1978*). Assuming $s$ is small and thus approximately additive, this recurrent process of fitness decay occurs roughly when the following inequality is satisfied (see Appendix 1, Section 1.2; *Neher and Shraiman, 2012*):

$$2sN_e\, e^{-\frac{U_N}{s}} << 1. \tag{5}$$

Given $U_N/s = 29$ as derived above, $N_e > 10^{15}$ is required to avoid continual deleterious mutation accumulation and fitness decline (*Figure 2b*). Thus, the purifying model requires levels of drift unrealistic at the within-person or across-globe scales. Simulations confirm that deleterious mutations accumulate and compromise the ability of the purifying model to explain empirical $d_N/d_S$ decay in reasonably finite populations (*Figure 2c*). Moreover, continuous accumulation of mutations in such populations decreases fitness so much that the average genome contains a sizable fraction (~10%) of deleterious alleles after 1 million years (*Figure 2c*), assuming $N_e = 10^9$ and one generation a day (*Korem et al., 2015*). Even if this decreased fitness was biologically maintainable, the accumulation of so many deleterious mutations would lead to many potential adaptive back mutations, complicating the efficiency of purifying selection. Consequently, this value of $U_N/s$ is simply incompatible with a model where a vast majority of alleles are already optimal.

Lastly, the intolerance of the purifying model to adaptation and transmission is particularly problematic. Within-host adaptive sweeps have been observed in *Bacteroides fragilis* (*Zhao et al., 2019*) and other *Bacteroides* (*Garud et al., 2019*). Such adaptation interferes with inefficient purifying selection; deleterious mutations are likely to hitchhike (*Desai et al., 2013*) to fixation on the genomic background of adaptive mutations. Any given weakly deleterious mutation with $s = 3.5 \times 10^{-5}$ cannot be purged from a within-host population on the timescale of human lifetime (assuming ~1 generation per day), and thus if any adaptive sweep occurred within that host, it would either hitchhike to fixation or be completely removed from the population. Similarly, deleterious mutations can also hitchhike to fixation during neutral transmission bottlenecks, thereby raising the average number of deleterious

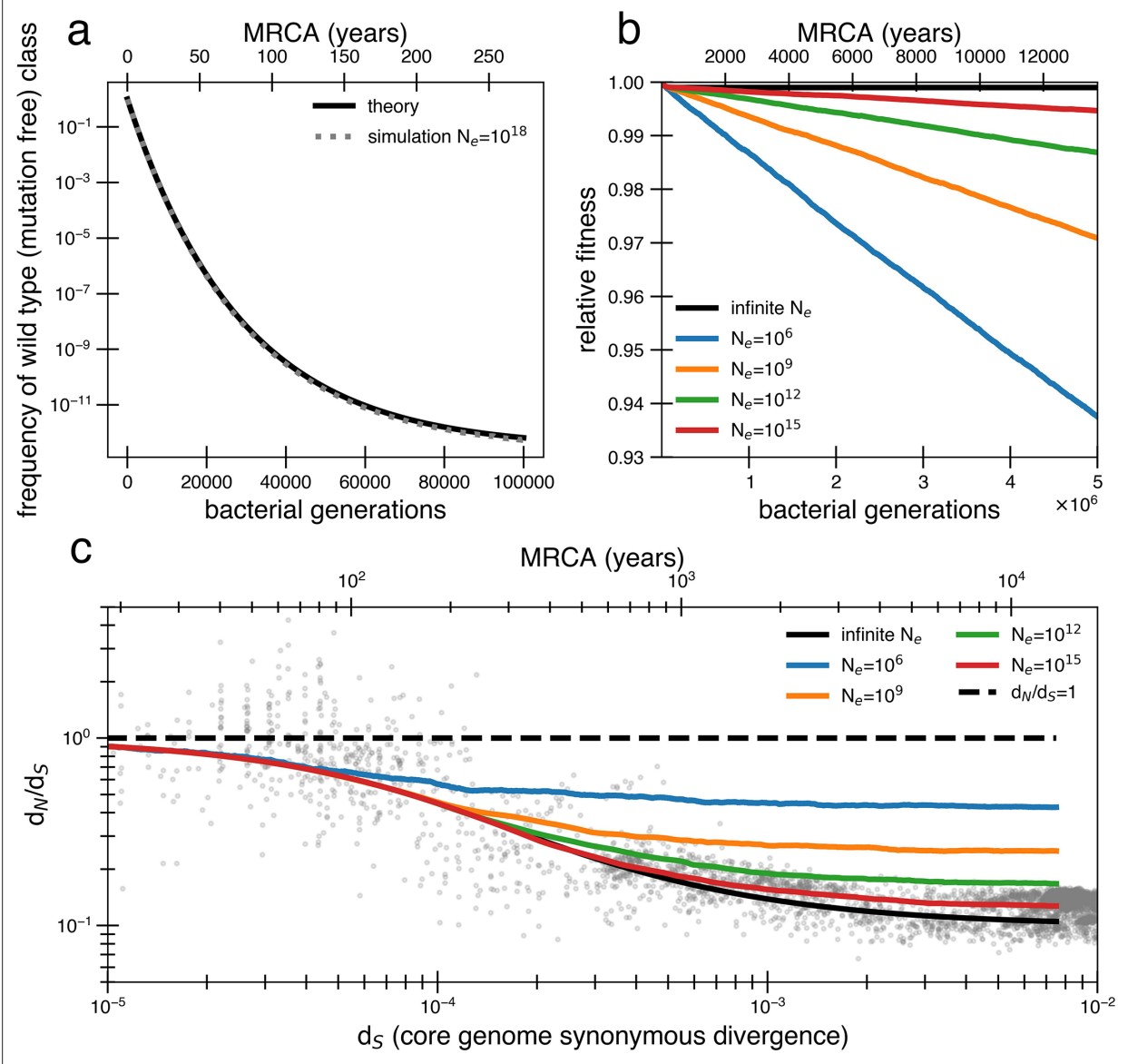

**Figure 2.** Models of extremely weak purifying selection that can fit the data suffer from mutation accumulation and fitness decay. (**a**) The temporal dynamics of the least-loaded class in a large population under the purifying selection model. The black line represents the predicted frequency of the wild-type (mutation-free) class over time. The simulation curve shows simulation results assuming constant purifying selection in an exceptionally large effective population size ($N_e = 10^{18}$; see text for a discussion of population size) under a slightly modified Wright–Fisher model ('Methods'). (**b**) As a consequence of the loss of the least-loaded class, fitness declines in finite populations over time. Colored lines indicate simulations from various effective population sizes with mutations of constant selective effect. The deleterious mutation rate in the simulation is $1.01 \times 10^{-3}$ per genome per generation. (**c**) Using the same simulations as in panel (**b**), we see that realistic global effective population sizes fail to fit the $d_N/d_S$ curve, with different asymptotes. The black line denotes the infinite population theoretical model, and the colored lines indicate increasing effective population sizes, which change the strength of genetic drift in the simulations. Larger values of $s$ and models in which all mutations are deleterious cannot fit the data (*Figure 2—figure supplement 1*, *Figure 2—figure supplement 2*). Generations are assumed to occur once every day.

The online version of this article includes the following figure supplement(s) for figure 2:

**Figure supplement 1.** Larger selective coefficients that can prevent mutation accumulation in simulations lead to less optimal data fit.

**Figure supplement 2.** Even assuming all mutations are deleterious suggests a higher asymptotic $d_N/d_S$.

**Figure supplement 3.** High rates of recombination are unlikely to rescue a model of weak purifying selection.

**Figure supplement 4.** Transmission bottlenecks and adaptation further complicate purifying selection.

mutations per cell in the population, furthering mutation accumulation, and hampering the efficiency of purifying selection. Simulations confirm that even infrequent adaptive sweeps and bottlenecks have tangible impacts on $d_N/d_S$, including raising the asymptote (*Figure 2—figure supplement 3*).

## Neither recombination nor differential selection at transmission can easily rescue a model of weak purifying selection

Homologous recombination, which occurs at detectable rates within human gut microbiomes and within the *Bacteroidales* order (*Liu and Good, 2024*), cannot rescue a population from Muller's ratchet when such weakly deleterious mutations are so frequent. If we assume a generously high rate of recombination, such that a mutated nucleotide is 500 times more likely to be reverted via recombination than mutation ($r/m$ = 500) (*Torrance et al., 2024*; *Liu and Good, 2024*) and brings along a single linked synonymous mutation during each recombination event, the decay of $d_N/d_S$ still cannot be recreated in a population of size $10^9$ and fitness will still decay (*Figure 2—figure supplement 4*). The inability of recombination to suppress mutation accumulation in this regime arises because the selective advantages themselves are still too small to sweep faster than the rate at which mutations accumulate. While recombination does allow $d_N/d_S$ to eventually decay, the rate of decay is much slower than observed, resulting in a poor fit to the data (*Figure 2—figure supplement 4*). While higher values of $N_e$ or a higher recombination rate could theoretically approximate the absence of linkage and escape of Muller's ratchet, we note that the maximum $r/m$ across bacteria is estimated to be <50 (*Torrance et al., 2024*) and our simulations are therefore conservative.

Our presentation so far has implicitly assumed that weak purifying selection has been acting continuously and that values of $s$ are constant for any given allele over time. However, apparently weak purifying selection might theoretically emerge from mutations that spend periods under neutral selection (or even local positive selection) and larger periods under strong negative selection, with the estimated value of $s$ reflecting the harmonic mean (*Culyba and Van Tyne, 2021*; *Loo et al., 2020*). However, such models will have a hard time overcoming mutation accumulation. For example, a model in which purifying selection acts only during transmission still cannot prevent mutation accumulation without unrealistic assumptions. In particular, the selection-at-transmission model would still require ~29 non-neutral mutations in the average adult population, which implies a very low frequency of the least-loaded class. Assuming each host's population gets replaced once every 10,000 bacterial generations (~26 years), such a model would require the least-loaded class to be 6000× more likely to colonize them than the average genotype in the population ($(1 + 10,000s)^{29}$). The presence of rare cells with strong selective advantages would suggest super-spreading across human microbiomes, which has yet to be reported in the human microbiome (*Faith et al., 2013*). More importantly, Muller's ratchet would still click because of the low frequency of this least-loaded class.

## Adaptive reversions can explain the decay of $d_N/d_S$

If purifying selection cannot explain the decay in neutral mutations, what can? One particularly attractive process that removes nonsynonymous mutations over time is strong adaptive mutation and subsequent strong adaptive reversion of the same nucleotide when conditions change. Such reversions are likely to sweep in large populations when mutations are adaptive locally but deleterious in other environments (*Ascensao et al., 2023*). In the gut microbiome, these alternative environments could represent different hosts (*Figure 3a*) or environmental changes within a single host (e.g., diet, medication, other microbes). As an illustrative example, the presence of a bacteriophage in one gut microbiome might select for a loss-of-function mutation (premature stop codon or otherwise) in a phage receptor, driving this mutation to fixation in its host, but reverting to the wild-type receptor when transmitted to a phage-free host. Reversions are most likely when compensatory mutations that counteract a mutation's deleterious effects are either scarce or not as beneficial as direct reversion (*Levin et al., 2000*) (i.e., provided a premature stop codon); we discuss models that include compensatory mutations later in this section.

Adaptive nonsense mutations have been observed to emerge frequently within individual people in both pathogens (*Culyba and Van Tyne, 2021*; *Lieberman et al., 2011*; *Key et al., 2023*; *Shopsin et al., 2008*) and commensals (*Zhao et al., 2019*; *Barreto et al., 2023*). Identifying reversions in vivo requires both high temporal resolution and deep surveillance such that the probability of persistence of ancestral genotype is removed (*Snitkin et al., 2013*) despite this difficulty, reversions of stop codons

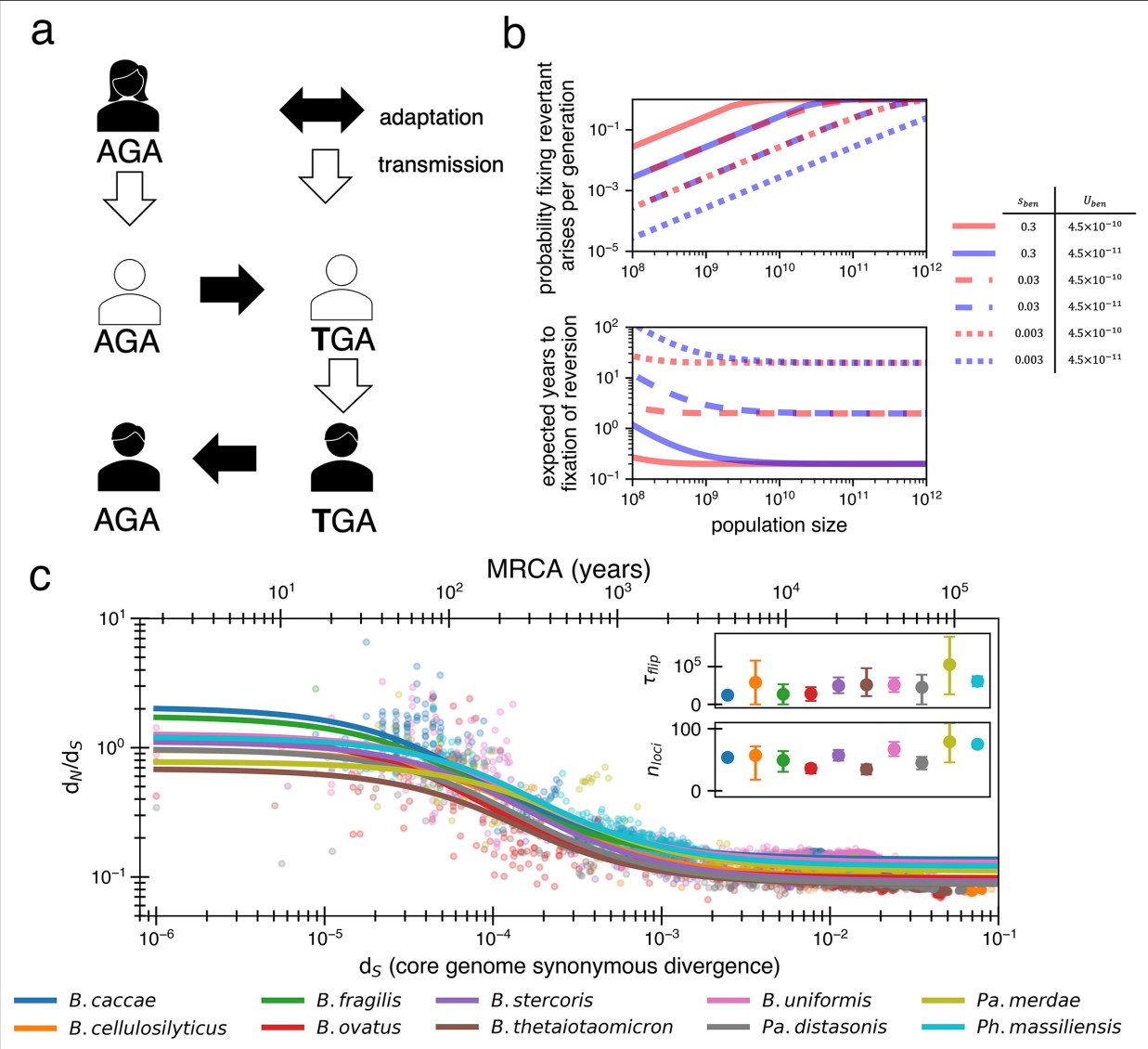

**Figure 3.** Locally adaptive mutations and subsequent reversions can explain the decay of nonsynonymous mutations. (**a**) Cartoon schematic depicting a potential reversion event within a single transmitted lineage of bacteria. The color of each individual indicates a different local adaptive pressure. Closed arrows represent mutation while open arrows indicate transmission. (**b**) Reversions become increasingly likely at larger population sizes and are nearly guaranteed to occur and fix within 1–10 years when strongly beneficial in gut microbiomes. The probability of revertant arising and fixing (top panel) is calculated as $1 - (1 - 2s_{ben}U_{rev})^{N_e}$, and the expected time to fixation of reversion (bottom panel) is calculated as $\frac{1}{2N_e s_{ben} U_{rev}} + \frac{ln(N_e)}{s_{ben}}$ ($U_{rev} = 4.5 \times 10^{-10}$ per generation). Generation times are assumed to be 1 day. Note that the mutation rate does not affect time to fixation much when $N_e$ is large. Here, we assume no clonal interference or bottlenecks, though simulations do take these processes into account. See Appendix 1, Section 2.2 for derivation. Each line type displays a different selective advantage coefficient. (**c**) The adaptive reversion model can fit the data. Each colored line shows the fit for a different species. The median $R^2$ = 0.82 (range 0.54–0.94). Fit minimizes logarithmic variance. See ***Figure 3—figure supplement 1*** for alternative fitting linear variance. See ***Figure 3—figure supplement 2*** for species individually. Insets: fit parameters for $\tau_{flip}$, the average number of generations for *a given* environmental pressure to switch directions and $n_{loci}$, the average number of sites under different fluctuating environmental pressures. The scale of the y-axis is linear. Confidence intervals are from 999 bootstrapped resamples.

The online version of this article includes the following figure supplement(s) for figure 3:

**Figure supplement 1.** Potential for even more adaptation if fitting nonlogarithmic $d_N/d_S$.

**Figure supplement 2.** Reversion model fits by individual species.

have been observed in mouse models (*Sousa et al., 2017*) and during an outbreak of a pathogen infecting the lungs of people with cystic fibrosis (*Poret et al., 2024*). While direct reversion has not yet been observed in gut microbiomes, premature stop codons are frequently observed. Among the 325 observed nonsynonymous de novo mutations in a study of within-host *B. fragilis* adaptation (*Zhao et al., 2019*), 28 were premature stop codons. This frequency is significantly higher than expected by chance (p=0.015; 'Methods'). Moreover, 4 of the 44 mutations in 16 genes shown to be under adaptive evolution on this short timescale were stop codons. These same 16 genes show a signature of purifying selection on long timescales (*Figure 1a*).

Traditionally, mutational reversions of stop codons and other mutations have been considered exceedingly unlikely and have been ignored in population genetics (*Tajima, 1996*), with a few exceptions (*Charlesworth and Eyre-Walker, 2007*). However, for a bacterial population within a human gut microbiome, the likelihood of a mutational reversion is quite high. A single species within the gut microbiome can have a census population size of $10^{13}$, with generation rates ranging from 1 to 10 per day (*Sender et al., 2016*; *Korem et al., 2015*). Taking a conservative estimate of one generation per day and a within-person $N_e$ of $10^{10}$ (e.g., bacteria at the end of the colon may not contribute much to the next generation; *Labavić et al., 2022*), reversions become highly probable (*Figure 3b*; see Appendix 1, Section 2.2). Given a mutation rate of $10^{-9}$ per site per generation, we anticipate 10 mutants at any given site each generation. In the large population sizes relevant for the gut microbiome, a beneficial mutation will then take substantially longer to sweep the population than occur, with values of $s_{ben} > 1\%$ generally sweeping within 10 years (*Figure 3b*). Consequently, if selection strongly benefits a reverting mutation, a genotype with a beneficial mutation is essentially guaranteed to emerge within days to weeks and replace its ancestors within the host within months to years.

Given its plausibility, we now consider if the reversion model can explain the observed decay of $d_N/d_S$. The dynamics of the reversion model can be given by

$$\frac{d\overline{N}_{transient}}{dt} = \frac{1}{\tau_{flip}} \left( n_{loci} - \overline{N}_{transient} \right) - \frac{1}{\tau_{flip}} \overline{N}_{transient}. \tag{6}$$

With the corresponding solution for $\overline{N}_{transient}$ being (see Appendix 1, Section 3.1)

$$\overline{N}_{transient}\left(t\right) = \frac{n_{loci}}{2} \left( 1 - e^{-\frac{2t}{\tau_{flip}}} \right). \tag{7}$$

Here, $n_{loci}$ denotes the number of loci that experience distinct sources of fluctuating selection. The parameter $\tau_{flip}$ represents the average number of generations required for the sign of selection at a chosen locus to flip and determines the key point in the $d_N/d_S$ decay curve where $\overline{N}_{transient}\left(t\right)$ begins to drop. We note that a locus here could be a nucleotide, gene, or gene set – any contiguous or noncontiguous stretch of DNA in which two knockout mutations would be just as beneficial or harmful as one mutation. We again use α to represent the proportion of nonsynonymous mutations that are neutral. Using *Equation 7*, we obtain a formula for $d_N/d_S$ that has only three free parameters when a single value for $\mu_S$ is chosen:

$$\frac{d_N}{d_S} = \alpha + \frac{n_{loci}}{6\mu_S t} \left( 1 - e^{-\frac{2t}{\tau_{flip}}} \right). \tag{8}$$

When fitting the $d_N/d_S$ curve, the values obtained are reasonable in the context of bacterial genomics, with median best-fit values across species of $\tau_{flip} = 46,000$ bacterial generations (range 25,000–105,000) and $n_{loci} = 55$ (range 34–80). Given daily bacterial generations, this value of $\tau_{flip}$ suggests the sign of selection on a given allele would flip approximately every 110 years. The average time for any pressure to flip would thus be approximately every 2 years, or less frequently if adaptive events occur in bursts (e.g., upon transmission to a new host). While 55 loci under distinct selective pressures may seem high, *Bacteroidetes* genomes are known to have dozens of invertible promoters (up to 47 in *B. fragilis*; *Jiang et al., 2019*). Invertible promoters are restricted out of the genome and re-ligated in the opposite direction to turn gene expression on or off. The number of invertible

promoters in a given genome approximates a lower bound on the number of fluctuating selective pressures that these genomes frequently experience. Interestingly, adaptive loss-of-function mutations reported in *B. fragilis* affect the same genes regulated by invertible promoters (*Zhao et al., 2019*). The plausibility of these fit parameters lends support to a model in which $d_N/d_S$ decays solely based on strong and recurrent local adaptations.

To ensure a reversion model is robust to finite populations, we performed simulations using fit parameters. These simulations capture the dynamics of a single population evolving as it transmits across a series of hosts through random bottlenecks (*Figure 4a*; 'Methods'); these simulations allow for clonal interference between adaptive mutations. We allow new pressures to arise independent of bottlenecks as new selective forces (phage migration [*Koskella and Brockhurst, 2014*]; immune pressures [*Barroso-Batista et al., 2015*]; dietary changes [*Carmody et al., 2019*]) can emerge throughout the lifespan and independent of migration; forcing transmission and bottlenecks to coincide gives similar results (*Figure 4—figure supplement 1*). As in the purifying selection simulations, the per base pair mutation rate is $10^{-9}$, and 90% of nonsynonymous substitutions are deleterious, but this time they have a larger *s* of 0.003 (*Robert et al., 2018*) and are thus purged more quickly from the population. Notably, while some of these deleterious mutations hitchhike to fixation during bottlenecks and adaptive sweeps, fitness does not decay because these mutations are subsequently reverted with adaptive sweeps (*Figure 4—figure supplement 2*). If deleterious mutations had significantly smaller *s*, they would be unable to be reverted due to the long time needed to reach fixation, even if bottlenecks and adaptive events are less frequent (*Figure 2—figure supplement 3*).

We note that other complex models that include reversion and other processes are also possible. For example, a model with a very large number of loci with selective tradeoffs and pressures that act only transiently (nonfluctuating) could potentially fit the data. However, the agreement between $n_{loci}$ and the number of invertible promoters, and the finding of parallel evolution in vivo, suggests the fluctuating selection model is more realistic than a very many-sites model.

So far, we have assumed that only exact reversions are selected upon when the sign of selection returns to its original state. However, the reversion model can also accommodate compensatory mutations that exclude any selective advantage for reversion; these compensatory mutations can also be subject to reversion themselves. We conceptualize this as a random walk, in which a locus at a nonancestral state acquires a compensatory mutation with probability *p* or obtains a true reversion with probability $1 - p$ (see Appendix 1, Section 3.2). As long as $p \leq 0.5$, $d_N/d_S$ will decay to the same asymptote despite adaptive dynamics occurring. While compensatory mutations shift the timing of $d_N/d_S$ decay to the right, it can be shifted backward by decreasing $n_{loci}$ (*Figure 4—figure supplement 3a*). The condition $p \leq 0.5$ is easily met when $s_{ben} = 0.03$, until excluding compensatory mutations are 10 times more likely than true reversion and provide 95% of the selective advantage of the true reversion (*Figure 4—figure supplement 3b*). If selective pressures are stronger (as they might be in the presence of phage), true reversions will outcompete compensatory mutations even if the supply of compensatory mutations is greater or such mutations provide better relative compensation.

A critical consequence of the reversion model is that apparent and actual $d_N/d_S$ values diverge quickly. Even when the true genome-wide $d_N/d_S$ exceeds 1 – meaning that adaptive sweeps have been a dominant force in shaping genomes – the observed value can be close to 0.1 on long timescales. This disparity complicates the interpretation of $d_N/d_S$ as it becomes challenging to determine whether a genome or gene lacks nonsynonymous mutations due to reversions or negative selection. We confirmed the inability to detect adaptive selection on a gene when reversion is rampant by simulating protein phylogenies; even the advanced software PAML (Phylogenetic Analysis by Maximum Likelihood) (*Yang, 2007*) significantly underestimates actual $d_N/d_S$ (*Figure 4b*; 'Methods'). Without sufficient temporal sampling, no software can realistically estimate these hidden, adaptively driven nonsynonymous mutations.

Lastly, we sought to find evidence of past reversions of stop codons in certain genes by analyzing codon usage. Both leucine and serine have the property that they can be encoded by six codons, only one of which is highly stop codon adjacent (TTA for leucine and TCA for serine). Across the *B. fragilis* genome, these codons are depleted overall (13.48% usage rather than the neutral expectation of 16.67%). However, specific Clusters of Orthologous Genes (COG) categories are enriched in TTA and TCA codons relative to this baseline, including genes associated with transcription and cell envelope biogenesis (*Figure 4c*, 'Methods', *Supplementary file 1*). Further, when functionally annotated genes

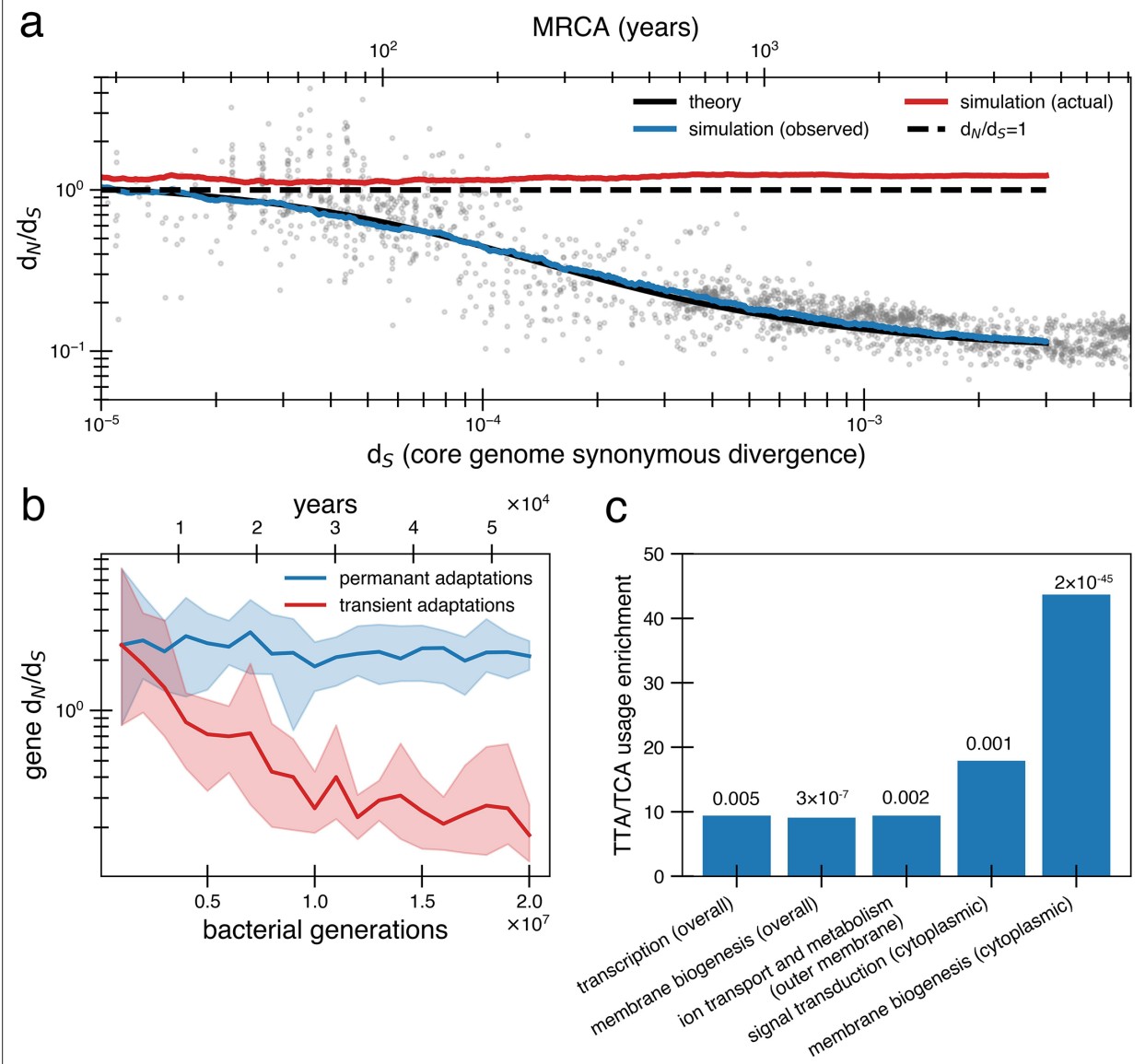

**Figure 4.** Under a model of reversion, the apparent $d_N/d_S$ on long timescales underestimates the extent of adaptive evolution. (**a**) The reversion model successfully fits the data in simulations. We simulate a population of size $10^{10}$ that has a bottleneck to size 10 on average every 10,000 generations (~27 years or a human generation [**Wang et al., 2023**] given a bacterial generation a day), with one adaptive pressure ($s_{ben} = 0.03$) occurring on average every 840 generations independently of bottlenecks (see **Figure 4—figure supplement 1** for an alternative where bottlenecks and selection are correlated). New pressures either require forward mutations (which can be acquired at a rate of $1.1 \times 10^{-8}$ per available locus per generation) or reversions (which can be acquired at a rate of $4.5 \times 10^{-10}$ per available locus per generation), the balance of which depends on the history of pressures on the tracked genome (i.e., more past forward pressures implies more potential future reverse pressures). Based on the best fit to the data, we use $n_{loci} = 55$. Deleterious mutations occur at a rate of $1.01 \times 10^{-3}$ mutations per genome and have $s = 0.003$ and can themselves be reverted. More details on the simulation can be found in 'Methods'. Each curve represents the average of 10 runs; the blue line shows the observed pairwise $d_N/d_S$ while the red line includes adaptive mutations and reversions. The theory line is the result of **Equation 8**. Observable $d_N/d_S$ decays because of reversion, while the actual $d_N/d_S$ of mutations that occurred is >1 when taking into account both forward and reverse mutations. (**b**) PAML (**Yang, 2007**) cannot detect true $d_N/d_S$ in a given gene in the presence of adaptive reversions. Both lines are $d_N/d_S$ as calculated by PAML on a simulated gene phylogeny. In the permanent adaptations simulation (blue), adaptive mutations are acquired simply and permanently. In the transient adaptations simulation (red), only more recent mutations will be visible while older mutations are obscured ('Methods'). Line is the average of 10 simulated phylogenies and shaded regions show the range. (**c**) Categories of genes in the *Bacteroides fragilis* genome (NCTC_9343) enriched for stop-codon adjacent codons (TTA and TCA) relative to the expectation from the rest of the genome ('Methods'). The use of these codons suggests these sequences may have recently had premature stop codon mutations. p-Values are displayed above bars and were calculated using a one-proportion Z-test with Bonferroni correction. See 'Methods', **Supplementary file 1**, and **Supplementary file 2** for more details.

The online version of this article includes the following figure supplement(s) for figure 4:

*Figure 4 continued on next page*

*Figure 4 continued*

**Figure supplement 1.** Effect of correlating mutations with bottleneck has little impact on the fit.

**Figure supplement 2.** Deleterious hitchhikers are reverted over time, preventing fitness decay.

**Figure supplement 3.** Inclusion of compensatory mutation in the reversion model shows that $d_N/d_S$ still decays provided reversion occurs at reasonable rates.

are further categorized by cellular localization, more gene categories exhibit enrichment (*Figure 4c*, 'Methods', *Supplementary file 1*), most notably genes involved in inorganic ion transport and metabolism that are localized to the outer membrane. Genes implicated in within-host *B. fragilis* adaptation (*Zhao et al., 2019*) are also found disproportionately in this category of outer membrane transporters (p=1.22 × 10⁻⁴; 'Methods', *Supplementary file 1*). Both the cell envelope and membrane-bound transporters are known to mediate interactions with the immune system and phage (*Sukhithasri et al., 2013*; *Ongenae et al., 2022*), and are therefore expected to experience fluctuating selective pressures. The enrichment of stop-codon adjacent codons in pathways associated with environment-dependent costs further supports a model in which adaptive mutational reversions are frequent.

## Discussion

In this study, we present a new interpretation of the time-dependent changes in $d_N/d_S$ for bacterial populations. We show that the traditional weak purifying selection model struggles to replicate theoretical results in realistic population sizes and propose an alternative model with opposite implications that are supported by analytical, simulation, and genomic results. Together, these results challenge the conventional view that high $d_N/d_S$ values on short timescales are an artifact and should not be trusted. Instead, the success of the reversion model suggests that adaptive dynamics are underestimated on long timescales because of the saturation of $d_N$.

It is perhaps not surprising that reversions have been relatively overlooked in previous literature. First, most population genetics theory focuses on eukaryotic organisms with smaller population sizes and longer generation times, for which reversion is less likely. The low likelihood of reversion in these populations has inspired the use of the convenient infinite-site model (*Tajima, 1996*), which assumes that reversions never occur and simplifies derivations. While smaller values of $N_e$ can be appropriate for modeling global bacterial dynamics – because bottlenecks and geography limit how many organisms effectively compete – they are inappropriate for within-gut populations, which are less structured. While gut microbiomes do have a spatial structure that reduces competition, theoretical work modeling this biogeography suggests that the census and active population sizes differ only approximately tenfold (*Ghosh and Good, 2022*; *Labavić et al., 2022*). This brings the within-gut microbiome $N_e$ to substantially larger than the per-nucleotide mutation rate, invalidating the infinite sites model. Secondly, while bacterial geneticists have long observed adaptive loss-of-function mutations, two common misinterpretations of population genetic parameters can underestimate the probability of reversion: molecular clock rates (μ), which are generally low, can easily be confused with the supply of potential mutations ($\mu N_e$) (*Lieberman, 2022*); and classical approaches that assess $N_e$ from genetic diversity vastly underestimate the currently active population size, particularly if a bottleneck recently occurred (e.g., during transmission). Lastly, simulating large populations, even when appropriate, is computationally difficult. As a consequence, population genetics simulations, including those of bacteria, have used relatively small population sizes (≤10⁶ organisms). We overcome computational limitations by tracking genetic classes rather than individual genotypes ('Methods'). While our approach does not allow explicit comparison between individuals within a population, we believe this framework represents a powerful method to simulate large population sizes when applicable.

Whether or not a reversion model can be applied beyond host-associated microbial populations remains to be explored. We only analyze microbiome data here, but we anticipate that analyses of highly curated $d_N/d_S$ decay curves from microbial pathogens could yield similarly plausible parameter fits for the reversion model given past observations of $d_N/d_S$ decay (*Rocha et al., 2006*). When effective population sizes are smaller than 10⁹, reversions are relatively unlikely. For example, while adaptive reversions can sweep individual gut microbiomes, we do not propose that reversions sweep the global bacterial population. Regardless, theoretical work on animal populations has shown that adaptive reversions are possible after local population bottlenecks (*Charlesworth and Eyre-Walker,*

*2007*). Similarly, environmental variations that change more rapidly than the timescale required for a local selective sweep (e.g., those imposed by daily dietary changes in the gut; or imposed by light-dark cycles in the environment) would be less likely to drive fixation and subsequent reversion than the less rapid changes considered here (e.g., phage migration) (*Cvijović et al., 2015*). On the other hand, adaptive reversions may be particularly relevant for viral populations, which are known to undergo within-host adaptation, have very large population sizes, and experience frequent bottlenecks (*Feder et al., 2017*). Reversions have commonly been observed in certain regions of the HIV genome and have been postulated to diminish measured substitution rates in those regions (*Druelle and Neher, 2023*).

Despite the success of a model of reversion alone in explaining $d_N/d_S$ decay, it remains possible that other forces could additionally contribute. While we have shown that purifying selection alone, either continuously or during transmission, cannot explain $d_N/d_S$ decay alone, it is possible that some degree of purifying selection could act alongside a reversion model. Similarly, directional selection could be incorporated into the reversion model by adjusting the parameter $\alpha$. While the true contribution of adaptive evolution to $\alpha$ is likely nonzero, it is difficult to fit with available data and it is therefore left for future work.

While we have presented evidence that recombination alone is unlikely to rescue a model of weak purifying selection, it remains possible that recombination could be included in a model that includes adaptation and, notably, could drive adaptive reversions. Microbial geneticists have frequently observed that recombined regions exhibit lower $d_N/d_S$ values compared to non-recombined regions (*Castillo-Ramírez et al., 2011*), a signature consistent with having already experienced reversion or purifying selection. Recombination could potentially revert multiple mutations at specific loci simultaneously, which might be particularly beneficial in the presence of genomic epistasis. Thus, despite the success of the mutation-driven model, it is likely that recombination plays some role in the decay of $d_N/d_S$.

While more direct observation of adaptive reversions is currently lacking, we propose that this paucity is simply an artifact of lacking samples along a line of descent with sufficient genomic resolution. Despite this challenge in observation, a recent study tracking de novo mutations between mothers and infants revealed several cases of apparent reversion, with elevated values of $d_N/d_S > 1$, though not significantly so (*Chen and Garud, 2022*). Moreover, many short-term studies in the gut microbiome and beyond have revealed strong evidence of within-person adaptation, including parallel evolution (*Lieberman et al., 2011*; *Marvig et al., 2015*; *Zhao et al., 2019*; *Cooper and Lenski, 2000*) and loss-of-function changes like premature stop codons (*Key et al., 2023*) – with low long-term $d_N/d_S$ values in these same short-term genes (*Vigué and Tenaillon, 2023*). We note that adaptation and reversion do not result in parallel evolution in the genomic record if various initial mutations result in the same phenotype (i.e., loss-of-function mutations); however, it would result in changes in codon usage bias we have shown (*Figure 4c*).

The shortcomings of the purifying model and the success of the reversion model under realistic assumptions highlight the importance of studying evolution in real time for understanding evolutionary dynamics. In addition, our results emphasize the importance of simulating large population sizes for explaining observations in bacterial population genomics, spotlight the potential for strong adaptation in bacterial populations, and underscore the need for continued development of population genetics theory for microbial populations.

## Methods

### Data and parameter estimation

Data was obtained from *Shoemaker et al., 2022* and was initially generated by *Garud et al., 2019*. Pairwise $d_N/d_S$ values can be found in the GitHub repository. The parameters are estimated using scipy.optimize.curve_fit. The fit minimizes the RMSD of the *logarithmic* $d_N/d_S$. If we fit the data by minimizing just $d_N/d_S$ on a linear scale, we get $s \approx 2.0 \times 10^{-5}$, which suggests an even weaker purifying selection (*Figure 1—figure supplement 1*). We analyzed the 10 species with the most data points reflecting short divergence times ($d_S < 0.0005$), which is critical for data fit.

### Population simulations overview

The majority of the computational simulations performed are built upon the idea of the Wright–Fisher model with selection (*Tataru et al., 2017*) that population generations can be determined from a

multinomial distribution. However, we have made some changes to generalize this model for our purposes.

First, the simulations do not necessarily assume a constant population size but rather assume the population grows via a logistic growth model with a capacity to allow for the implementation of bottlenecks. Specifically, if $P[t]$ is the population on generation $t$ and $K$ is the population capacity, then

$$\overline{P[t+1]} = P[t] + P[t]\left(1 - \frac{P[t]}{K}\right)$$

And

$$P[t+1] = Poiss(\overline{P[t+1]}).$$

The population size is a Poisson random variable as we choose to determine the offspring of individual genetic classes as a Poisson random variable. We note that except for the very first few generations and after bottlenecks, the population size only has small fluctuations around a fixed capacity.

To speed the simulation up and enable the simulation of very large population sizes, we implemented a variety of genotype classes, rather than tracking each genotype individually. Genotype classes are similar to the practice of simulating fitness classes (**Desai and Fisher, 2007**), though we manage the number of unique classes via Poisson merging and splitting.

For all simulations, we start from a single organism that begins with 500,000 neutral alleles, representing a core genome size of this many codons that has yet to receive any mutations. When a mutation occurs, one allele may change types or stay the same, depending on the mutation received and the state of the randomly chosen codon. For example, a deleterious mutation occurring at a codon already in a deleterious state does not change the genotype class of the organism.

The specific implementations and additional parameters used for this model are provided in the following sections. Here, we outline the theory that ensures that genotype classes accurately represent such a population and enable the calculation of fitness. Consider the total population of size, $P[t]$, at generation $t$, as a composite of multiple different classes. The number of individuals in class $j$ on generation $t$ will be $A_j[t]$. We have

$$P[t] = \sum_{j \geq 0} A_j[t].$$

Within class $j$, we store several variables that provide information about the genotype of members of $A_j[t]$. Specifically, we store a number $j_k$ that specifies the number of alleles of type $k$ in the class $j$. Examples of potential types that are used in our work include deleterious alleles, adaptive alleles, and alleles that result from reversion. Each type of allele is associated with a specific selective advantage $s_k$. We can now write a formula to calculate the absolute fitness $F_j$ of class $j$:

$$F_j = \prod_{k \geq 0} \left(1 + s_k\right)^{j_k}.$$

From the absolute fitness $F_j$, we calculate the average absolute fitness of the population on generation $t$ via

$$E[F] = \frac{\sum_{j \geq 0} F_j A_j[t]}{P[t]}.$$

We now calculate the relative fitness of class $j$ on generation $t$ as

$$f_j = \frac{F_j}{E[F]}.$$

Next, we calculate the expected size of class $j$ in the next generation with

$$A_j\,[t+1] = f_j\left(A_j\,[t] + A_j\,[t]\left(1 - \frac{P\,[t]}{K}\right)\right).$$

Note that

$$P\,[t+1] = \sum_{j \geq 0} A_j\,[t+1] = \sum_{j \geq 0} f_j\left(A_j\,[t] + A_j\,[t]\left(1 - \frac{P\,[t]}{K}\right)\right) =$$

$$\sum_{j \geq 0} \frac{F_j}{E\,[F]}\left(A_j\,[t] + A_j\,[t]\left(1 - \frac{P\,[t]}{K}\right)\right) = P\,[t] + P\,[t]\left(1 - \frac{P\,[t]}{K}\right).$$

This allows us to use a logistic model of growth to represent population size rather than being constrained to fixing it, which is useful for simulating bottlenecks.

To account for genetic drift through random fluctuations, we rewrite the above equations to be

$$A_j\,[t+1] = Poiss\left(f_j\left(A_j\,[t] + A_j\,[t]\left(1 - \frac{P\,[t]}{K}\right)\right)\right).$$

which also implies

$$P\,[t+1] = Poiss\left(P\,[t] + P\,[t]\left(1 - \frac{P\,[t]}{K}\right)\right).$$

Note that this simulation still has equivalent dynamics of the frequencies of classes as a Wright–Fisher model with selection in the case of a fixed population size due to the ability to split Poisson processes, that is,

$$Pr\left(A_j\,[t+1] = X | P\,[t+1]\right) = Pr\left(Bin\left(P\,[t+1], \frac{f_j A_j\,[t+1]}{P\,[t]}\right) = X\right).$$

Mutations are added in every generation depending on the mutation rate. Only single mutants are generated per generation, and an organism cannot get more than one mutation per generation. The number of new mutants is determined by the binomial distribution. New mutants are added then to their appropriate class $j$. For example, if a deleterious mutation is gained in a class with 10 deleterious alleles (and nothing else), this new mutant will increase the population size of the class with 11 deleterious alleles (and nothing else) while decreasing the population size of the class with 10 deleterious alleles (and nothing else).

By grouping individuals in classes rather than by genotype, computational costs can be greatly cut down. Grouping individuals does not affect the dynamics of the simulation because the merging of Poisson processes is still Poisson. The downside to this approach is information loss, though by designing custom alleles, we can track specific mutational histories like reversions.

## Purifying selection simulations

The purifying selection simulations (*Figure 2*) utilized the base framework as mentioned above. Effective population sizes ($N_e$) varied from $10^6$ to $10^{18}$ depending on the simulation. The simulation begins with an initial organism with 500,000 neutral alleles (representing a WT core genome). The population quickly grows to the carrying capacity ($N_e$) and follows logistic growth (see 'Population simulations overview'). The deleterious nonsynonymous mutation rate per genome per generation is $1.01 \times 10^{-3}$. These deleterious mutations have a selective disadvantage of $s \approx 3.5 \times 10^{-5}$. Synonymous and neutral nonsynonymous mutations are not simulated directly as they are neutral and are instead assumed to accumulate in the population with an average rate of $3.75 \times 10^{-4}$ and $1.12 \times 10^{-4}$ per genome per generation, respectively.

We estimate the average $d_N/d_S$ of the population by taking the average number of codon differences between two individuals in the population to be twice the average number of mutations in the population. This approximation is valid due to the lack of selective sweeps, bottlenecks, and

large effective population size, which results in expected coalescent time between random individuals being $10^6$–$10^{18}$ generations (far longer than our simulations).

Variations of this basic purifying selection model are performed as described in the article, including increasing the mutation rate to $1.13 \times 10^{-3}$ mutations per genome per generation (*Figure 2—figure supplement 2*) and the simulation of recombination (*Figure 2—figure supplement 3*). For simulations of recombination, we assume that transitions to the ancestral state (purging the deleterious allele) occur at a rate of $2.5 \times 10^{-7}$ per codon per genome per generation and bring along a synonymous mutation (tracked via the number of recombinations to the ancestral state). This procedure does not allow for recombination to purge multiple deleterious alleles at a time; such events are unlikely given that deleterious alleles are rare and randomly distributed. We also include a model in which synonymous mutations are not included in this reversion event.

The simulation of purifying selection through bottlenecks and infrequent adaptive sweeps was performed as in the modified version of the reversion model (see 'Reversion simulations'), though with less frequent adaptation and larger and less frequent bottlenecks.

## Stop codon enrichment in the Zhao and Lieberman et al. dataset

Table S7 in *Zhao et al., 2019* provides an Excel sheet detailing all observed mutations. There were 325 observed nonsynonymous mutations of which 28 were stop codons. Under a null model, there are 415 possible permutations of initial codon and codon one mutation (see 'Code availability') away that result in a nonsynonymous substitution of which 23 lead to a stop codon. Assuming no preference for specific mutation or initial codon, we would expect roughly 18 stop codons in this data. Under a null binomial distribution, the p-value for obtaining 28 or more is 0.015.

## Reversion simulations

We simulated gut bacterial populations using a modified Wright–Fisher model (see 'Population simulations overview') to monitor mutation acquisition over time compared to an ancestor. Like all simulations, we begin with a single organism with 500,000 neutral alleles to represent the WT core genome. The population can grow to a capacity of $10^{10}$ via a logistic growth model. Environmental changes occur with a probability of $\frac{n_{loci}}{\tau_{flip}}$ per generation, triggering an average of one selective pressure per environmental change, modeled by a Poisson distribution. Population bottlenecks to 10 individuals occur independently of environmental changes with a probability of $10^{-4}$ per generation (see *Figure 4—figure supplement 1* for an alternative in which bottlenecks and environmental change are correlated).

Both adaptive selective pressures and adaptive mutations are categorized into two allele types: forward and reverse. These classes are designed to enable tracking of complete mutational history and therefore recorded relative to the ancestral state rather than the current state. Thus, actual $d_N/d_S$ is calculated as the sum of these mutations and observed $d_N/d_S$ using their difference (plus asymptomatic $d_N/d_S$). When releasing beneficial selective pressures, their classification as forward or reverse is based on the balance of previously released selective pressures: the probability of an adaptive pressure being classified as a reverse adaptation increases as the number of forward pressures increases and is equal to the difference between the forward pressures previously released ($q_F$) and reverse pressures previously released ($q_R$) divided by the number of loci (i.e., $\frac{q_F - q_R}{n_{loci}}$). All beneficial mutations have a selective advantage of $s_{ben} = 0.03$ (for forward or reverse). The rate at which mutations occur given an available pressure depends on whether the mutation is adapting to a forward or reverse pressure: the reversion rate is set at one-fifth the rate of the nonsynonymous per codon mutation rate ($4.5 \times 10^{-10}$ per generation per cell), while forward mutations are set at a rate five times higher than the nonsynonymous mutation rate because they can happen at multiple sites ($25\times$ the reversion rate; $1.1 \times 10^{-8}$ per generation per cell).

This simulation treats each adaptive mutation as occupying a unique codon in the core genome for simplicity. This assumes that the ancestral allele at a given locus has been purged before the next environmental change affecting that locus (or selective pressure); as theory suggests that a beneficial mutation takes 768 generations to fix (see Appendix 1, Section 2.2), compared to 46,200 generations for pressure shifts at any locus we believe this assumption is reasonable. To confirm this theory still holds in the presence of bottlenecks and clonal interference, we tracked the average number of beneficial mutations in the population relative to the number of selective pressures released at any

generation in simulations; we found only a 2% deviation between the average and expected total beneficial mutations over $2 \times 10^6$ generations.

Throughout the simulation, deleterious mutations occur at a rate of $1.01 \times 10^{-3}$ mutations per genome per generation, with a selective disadvantage of $s = 0.003$, and can be reverted to a deleterious reversion allele class (separate from the adaptive reversion allele class).

To calculate $d_N/d_S$, we assume the simulated population could be compared to an equivalent population but with distinct mutations, allowing us to calculate the $d_N/d_S$ as using double the current observed substitutions.

We assume during the reversion simulations that the ancestor has no initial transient mutations. We make this assumption for computational simplicity but the theoretical curve is equivalent whether starting from no revertible mutations or the equilibrium where half of the loci currently have forward mutations (assuming forward and reverse mutations occur at equal rates; see Appendix 1, Section 3.3).

## Testing standard $d_N/d_S$ software

We simulated gene sequences with selective pressures acting at specific sites for *Figure 4b*. For each genomic distance investigated (every 500,000 bacterial generations), we ran 10 simulations as described below, with each simulation resulting in 10 sequences derived from a branching process. In the permanent adaptations simulation (blue), adaptive mutations in the phylogeny are acquired simply and permanently. In the transient adaptations simulation (red), only more recent mutations in the same phylogeny will be visible while older mutations are obscured by reversion. Both sets of sequences were then fed to PAML v4.8 (*Yang, 2007*) for the estimation of $d_N/d_S$ values. PAML uses maximum likelihood analysis to estimate the rate of substitution that best explains a given phylogenetic tree.

For each simulation, we generated a random 1500 bp open-reading frame and designated 10% of codons as neutral, 10% under positive selection, and 80% under purifying selection. We introduced mutations and branches across several cycles, with each cycle representing 100,000 generations. For each cycle, we assigned mutations at random according to the following probabilities: 67.5% that a nonsynonymous mutation occurred at a codon under positive selection, 12.5% for synonymous mutation at any codon, 3.75% for nonsynonymous mutation at a nonselected site (neutral), and 16.25% for no mutation. These values were selected to give a $d_N/d_S$ of around 2 and to match the general ratios in the reversion model.

Two phylogenies were constructed from each simulation: both received identical mutations, but they differed in how nonsynonymous mutations at selective codon sites were visible at the end of the simulation. In the transient adaptations version of the phylogeny, nonsynonymous mutations at selective codon sites were reverted at the end of the simulation, except those that occurred within the last 500,000 generations. Reverted sites were converted to a synonymous substitution at a frequency based on the codon table (assuming an equal probability of all nucleotide mutations). Both versions of the sequences underwent multiple sequence alignment and neighbor-joining tree construction (Biopython; *Cock et al., 2009*). We calculated treewide $d_N/d_S$ ratios using PAML v4.8's codeML feature (*Álvarez-Carretero et al., 2023*), employing the M2a model to analyze site-specific selection.

## Closeness to stop codons

To evaluate possible enrichment for stop codon adjacency, we focused on TTA and TCA codons. TTA and TCA are ideal for measuring the likelihood of nonsense mutations because each has two point mutations that yield a stop codon, unlike the five other redundant codons encoding for the same amino acids (for both leucine and serine, one codon is singly stop codon adjacent and the last four are not stop codon adjacent). In *B. fragilis,* these codons have a codon usage rate of 13% for leucine and 14% for serine.

We annotated the reference genome NCTC_9343 with Bakta v1.9 (*Schwengers et al., 2021*) and obtained COG categories for each gene using eggNOG v5.0 (*Cantalapiedra et al., 2021*). Genes that did not have a functional COG category (35%) were removed. To control for unusual outlier genes skewing results, only the 15 COG groups that had at least 50 genes were considered for enrichment analyses. For each COG category, we calculated a null codon usage proportion based on the proportion of leucine and serine codons and compared this to the actual proportion using a one-proportion $Z$ test. To address the fact that genes in the same functional category but localized to different parts

of the cell may be under different selective pressures, we analyzed cellular location classifications from PSORTb v3.02 (*Yu et al., 2010*) and categorized genes by the combination of function and localization. We analyzed the 15 function-location combinations with more than 50 genes. After identifying those categories that were significantly enriched, we cross-referenced which categories of genes shown to be under adaptive within-person evolution in a previous study of *B. fragilis* within-person evolution were in *Zhao et al., 2019*. Of the 16 genes reported in that paper, 8 were assigned a functional COG category/cellular location and in the reference genome (NCTC_9343). Four of these were in outer membrane inorganic ion transport and metabolism, a significant enrichment (p = $1.22 \times 10^{-4}$; binomial test) (*Supplementary file 1*, *Supplementary file 2*).

## Code availability

Code and simulation results are available at https://github.com/PaulTorrillo/Microbiome_Reversions (copy archived at *Torrillo, 2023*).

## Acknowledgements

We thank Daniel Fisher, two anonymous reviewers, Benjamin Good, Erik van Nimwegen, and all members of the Lieberman Lab for their thoughtful feedback on this manuscript. We also thank William Shoemaker for making the data used in this work easily accessible and for his feedback on the manuscript. This work was funded by a grant from the National Institutes of Health (1DP2GM140922-01 to TDL) and a fellowship for the National Sciences Foundation (to PAT).

## Additional information

### Funding

| Funder | Grant reference number | Author |
|---|---|---|
| National Institutes of Health | 1DP2GM140922-01 | Tami D Lieberman |
| National Science Foundation | Graduate Research Fellowship Program | Paul A Torrillo |

The funders had no role in study design, data collection and interpretation, or the decision to submit the work for publication.

### Author contributions

Paul A Torrillo, Conceptualization, Software, Formal analysis, Investigation, Visualization, Methodology, Writing – original draft; Tami D Lieberman, Conceptualization, Supervision, Funding acquisition, Investigation, Writing – original draft, Project administration

### Author ORCIDs

Paul A Torrillo https://orcid.org/0000-0002-4618-6061
Tami D Lieberman https://orcid.org/0000-0001-5430-3937

### Decision letter and Author response

Decision letter https://doi.org/10.7554/eLife.93146.sa1
Author response https://doi.org/10.7554/eLife.93146.sa2

## Additional files

### Supplementary files

• Supplementary file 1. Enrichment of stop adjacent codon usage in specific gene categories of *Bacteroides fragilis* NCTC_9343, related to *Figure 4c*. Results display TTA/TCA enrichment for statistically significant categories of genes in *B. fragilis* NCTC_9343. Loci from these categories and mutated in *Zhao et al., 2019* (Table S7) are noted in column J.

• Supplementary file 2. Genes assigned to COG categories in *B. fragilis* NCTC_9343 and their

enrichment of stop-adjacent codons, related to *Figure 4c*. List of genes assigned COG categories (via eggNOG) to be used to evaluate for closeness to stop codons. Cellular location is also given (if predicted) by PSORTb. Annotations are from Bakta.

• MDAR checklist

## Data availability

Code and results of simulations are available at Github repository https://github.com/PaulTorrillo/Microbiome_Reversions (copy archived at *Torrillo, 2023*).

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

## Appendix 1

### Supporting information

#### 1.1 d$_N$/d$_S$ theory

The following is meant to provide a more in-depth walkthrough of how one can build up and interpret the purifying selection model and its effect of time dependence on d$_N$/d$_S$. To be accessible to a wide audience and self-contained, we have included enough detail that most sections should be followable with basic knowledge of calculus.

Assume an infinite population of organisms. Consider the existence of $m$ classes of nonsynonymous mutations. The number of mutations of the $i$th class in the population is represented by the variable $\overline{N}_i$. Each class $\overline{N}_i$ has an associated mutation rate $U_i$ (mutations per genome per generation per unit time) and an associated selective disadvantage $s_i$ (mutation purification per unit time). In the purifying selection model, we assume that $s_0 = 0$ and $s_{i>0} > 0$. We assume that both the mutation rate and the selective disadvantage of each class remain constant throughout time. We assume this is the global population and hence no migration. We then have

$$d\overline{N}_i = U_i dt - s_i \overline{N}_i dt \qquad (S1)$$

Assuming $i > 0$, we can integrate

$$\frac{dN_i}{U_i - s_i \overline{N}_i} = dt.$$

Using u-substitution, we let $u = \mu_i - s_i \overline{N}_{i_i}$ so that $\frac{du}{d\overline{N}_i} = -s_i$ which implies $d\overline{N}_i = \frac{du}{-s_i}$ so that

$$\frac{du}{-s_i u} = dt.$$

Integrating both sides, we get

$$\frac{-\ln(u)}{s_i} = t + C.$$

where $C$ is the constant of integration. Continuing we have

$$u = e^{-s_i t - s_i C}$$

$$\mu_i - s_i \overline{N}_i = e^{-s_i t - s_i C}$$

$$\overline{N}_i = \frac{\mu_i}{s_i} - \frac{e^{-s_i t - s_i C}}{s_i}.$$

We can remove the constant of integration and instead replace it with the initial condition $\overline{N}_i(0)$

$$\overline{N}_i(0) = \frac{U_i}{s_i} - \frac{e^{-s_i C}}{s_i}.$$

So, then we have that

$$C = \frac{\ln\left(U_i - s_i \overline{N}_i(0)\right)}{-s_i}.$$

So that we get

$$\overline{N}_i = \frac{U_i}{s_i}\left(1 - e^{-s_i t}\right) + \overline{N}_i(0) e^{-s_i t}.$$

Since $\overline{N}_i(0)$ should occur in a homogeneous population that is the most recent common ancestor of the individuals in the population we are observing, we assume $\overline{N}_i(0) = 0$, so we have

$$\overline{N}_i = \frac{U_i}{s_i}\left(1 - e^{-s_i t}\right).$$ (S2)

If $i = 0$, we have

$$\overline{N}_0 = U_0 t.$$

We also assume that there are synonymous mutations $\overline{S}$ that are neutral and occur with new mutations per unit time $U_S$ so that

$$\overline{S} = U_S t.$$

So, if we want to find $\overline{N}$ (the total number of nonsynonymous mutations), that will be given by

$$\overline{N} = U_0 t + \sum_{i \geq 1} \frac{U_i}{s_i}\left(1 - e^{-s_i t}\right).$$ (S3)

Now observe the following with $G$ as the number of base pairs in the core genome, then $d_N = 2\overline{N}/(G \times 3/4)$ and $d_S = 2\overline{S}/(G \times 1/4)$. The 2 comes from the fact that there are two diverged lineages when calculating d$_N$/d$_S$.

$$\frac{d_N}{d_S} = \frac{1}{3}\frac{\overline{N}/G}{\overline{S}/G} = \frac{U_0 t + \sum_{i \geq 1}\frac{U_i}{s_i}\left(1 - e^{-s_i t}\right)}{U_S t} = \frac{1}{3}\left(\frac{U_0}{U_S} + \sum_{i \geq 1}\frac{U_i/U_S}{s_i t}\left(1 - e^{-s_i t}\right)\right).$$ (S4)

The ⅓ is for normalization. While the above form is easier to analyze in terms of actual time, it should also be noted that data is given in terms of d$_S$ rather than $t$ so the following equivalent form can also be helpful when discussing fitting of the timescale dependence of d$_N$/d$_S$:

$$\frac{d_N}{d_S} = \frac{1}{3}\left(\frac{U_0}{U_S}\right) + \sum_{i \geq 1}\frac{2U_i}{Gs_i d_S}\left(1 - e^{-\frac{Gs_i}{2U_S}d_S}\right).$$

Furthermore, we can rewrite $U_i = 3\alpha_i U_S$ so that we have

$$\frac{d_N}{d_S} = \alpha_0 + \sum_{i \geq 1}\frac{2\alpha_i U_S}{Gs_i d_S}\left(1 - e^{-\frac{Gs_i}{2U_S}d_S}\right).$$

Note $\alpha_0$ is equivalent to $\alpha$ in the main text. Next, we can simply rewrite $\beta_i = \frac{Gs_i}{2U_S}$ so that we have

$$\frac{d_N}{d_S} = \alpha_0 + \sum_{i \geq 1}\frac{\alpha_i}{\beta_i d_S}\left(1 - e^{-\beta_i d_S}\right).$$ (S5)

To begin analyzing (S5), we consider the asymptotic behavior. First, we note that

$$\lim_{d_S \to +\infty} \alpha_0 + \sum_{i \geq 1}\frac{\alpha_i}{\beta_i d_S}\left(1 - e^{-\beta_i d_S}\right) = \alpha_0 = \frac{U_0}{U_S}.$$ (S6)

For initial behavior, we can use L'Hopital's rule

$$\lim_{d_S \to +0} \alpha_0 + \sum_{i \geq 1}\frac{\alpha_i}{\beta_i d_S}\left(1 - e^{-\beta_i d_S}\right) = \alpha_0 + \sum_{i \geq 1}\alpha_i = \frac{U_0 + \sum_{i \geq 1} U_i}{U_S}.$$ (S7)

Finally, we are interested in when d$_N$/d$_S$ ends up being in between these initial and final values. Each class $i$ has a different midpoint at which its contribution to d$_N$ /d$_S$ is a half. In mathematical terms, this can be summarized as

$$\frac{\alpha_i}{2} = \frac{\alpha_i}{\beta_i d_S}\left(1 - e^{-\beta_i d_S}\right).$$

which implies

$$\frac{\beta_i d_S}{2} = \left(1 - e^{-\beta_i d_S}\right).$$

Thus, we see that for any class of mutation, $\frac{1}{s_i}$ will determine the midpoint (assuming some constant $U_S$ and $G$). We can then imagine the curve as similar to a step function where the location of each step is determined by the corresponding $\frac{1}{s_i}$ and how far the function steps down will be determined by the size of the mutation rate $U_i$. Finally, we can surmise that the average step-down occurs at approximately the harmonic average

$$s^{-1} = \sum_{i \geq 1} \frac{U_i}{U_N s_i}.$$

where $U_N = \sum_{i \geq 1} U_i$, which represents the total non-neutral nonsynonymous mutation rate. Taking this all into account, here is exactly what is being fit in the $d_N/d_S$ curve. We are fitting the equation

$$\frac{d_N}{d_S} = \alpha + \frac{1 - \alpha}{\beta d_S}\left(1 - e^{-\beta d_S}\right). \tag{S8}$$

$\alpha$ is the fraction of nonsynonymous mutations that are neutral. Now $\beta$ is a compound parameter and we can fit $\beta = \frac{Gs}{2U_S}$ which is essentially half of the harmonic average of selective disadvantages in ratio with the synonymous mutation rate per site per generation. Now we can estimate the synonymous mutation per site per generation with the following:

$$10^{-9}\frac{mutations}{base\ pair\ \times\ generation} \times 3\frac{base\ pairs}{site} \times \frac{1}{4}\frac{synonymous\ mutation}{mutations} = \frac{3}{4} \times 10^{-9} \times \frac{synonymous\ mutations}{site\ \times\ generation}.$$

Thus, we are truly fitting

$$\frac{d_N}{d_S} = \alpha + \frac{\frac{3}{4} \times (1 - \alpha) \times 10^{-9}}{2sd_S}\left(1 - e^{-\frac{s}{\frac{3}{2} \times 10^{-9}}d_S}\right). \tag{S9}$$

## 1.2 Mutation accumulation

To analyze genetic drift and Muller's ratchet (*Haigh, 1978*; *Neher and Shraiman, 2012*), we will provide a brief overview of the approach well suited for our work. For any nonsynonymous mutation class $N_i$ with $i > 0$, we can track the size of a population with 0 mutations of class $N_i$, denoted by variable $W_i$, via

$$dW_i = s_i \overline{N}_i\left(t\right)W_i dt - U_i W_i dt. \tag{S10}$$

Here, $\overline{N}_i(t)$ is the average number of mutations of class $\overline{N}_i$ in the population and $s_i\overline{N}_i(t)W_i$ is equivalent to mean fitness. This equation reflects how every unit of time, the mutation-free class should increase by its selective advantage relative to the population though also loses members of the population to the mutation rate.

If we substitute in $\overline{N}_i(t)$, we get

$$dW_i = s_i\frac{U_i}{s_i}\left(1 - e^{-s_i t}\right)dt - U_i W_i dt.$$

$$dW_i = U_i\left(1 - e^{-s_i t}\right)W_i dt - U_i W_i dt.$$

$$dW_i = -U_i e^{-s_i t}W_i dt.$$

So then

$$\ln\left(W_i\right) = \frac{U_i}{s_i}e^{-s_i t} + C.$$

Assuming that $W_i$ at time 0 is given by $W_0$ (representing the initial population size which under our assumptions is always free of mutations and hence wild type), then

$$\ln\left(W_0\right) = \frac{U_i}{s_i} + C.$$

$$C = \ln\left(W_0\right) - \frac{U_i}{s_i}.$$

$$W_i\left(t\right) = W_0 e^{-\frac{U_i\left(1 - e^{-s_i t}\right)}{s_i}}. \tag{S11}$$

We see that if we set $W_0 = 1$, then everything can be given in terms of frequencies within the population. Asymptotically, we have that

$$\lim_{t \to \infty} W_0 e^{-\frac{U_i\left(1 - e^{-s_i t}\right)}{s_i}} = W_0 e^{-\frac{U_i}{s_i}}. \tag{S12}$$

Importantly, we also note the following: let $W$ be the frequency of the wild type (mutation-free class) and $W_0 = 1$. Then,

$$W\left(t\right) = \prod_{i>0} W_i\left(t\right) = e^{-\sum_{i>0}\frac{U_i\left(1 - e^{-s_i t}\right)}{s_i}}. \tag{S13}$$

which implies

$$\lim_{t \to \infty} e^{-\sum_{i>0}\frac{U_i\left(1 - e^{-s_i t}\right)}{s_i}} = e^{-\sum_{i>0}\frac{U_i}{s_i}} = e^{-\frac{U_N}{s}}. \tag{S14}$$

Once again using

$$U_N = \sum_{i\geq 1} U_i.$$

And

$$s^{-1} = \sum_{i\geq 1}\frac{U_i}{U_N s_i}.$$

Furthermore, the average time for the frequency in the population without mutations of class $i$ to reach one-half of the logarithm of the asymptotic frequency is the same when using the above simplifications. In other words, the difference in how much and how fast the wild type is lost should not depend too heavily on the distribution of selective coefficients.

Finally, we can predict if the least-loaded class will be lost to drift. We can form this prediction via the following. We assume that if the least-loaded class $W$ drops below its steady-state frequency, $e^{-\frac{U_N}{s}}$, it will have advantage $s$ every generation. If the least-loaded class has advantage $s$, then it has a $2s$ probability of extinction (see Appendix 1, Section 2.1). Hence, if we expect there to be $N_e\,e^{-\frac{U_N}{s}}$ individuals, we can estimate that mutation accumulation will occur when

$$2sN_e\,e^{-\frac{U_N}{s}} \ll 1. \tag{S15}$$

## 2.1. Extinction and fixation probability

Here, we will derive how the fixation probability of a mutation with selective advantage $s_{ben}$ is approximately $2s_{ben}$. This is a standard result that can be found in classic population genetics textbooks. It is usually derived via differential equations but can also be obtained more classically from the study of branching processes (**Haldane, 1927**) ,which we will use here. First, we assume that individuals reproduce via a Galton–Watson branching process with mean of $1 + s_{ben}$. We also assume there are no other mutations in the population that can interfere with fixation or extinction. A fundamental result from the study of branching processes is that the extinction probability is given by the smallest non-negative root of the branching processes corresponding probability generating function, $f(z)$. Being a Galton–Watson branching process, the probability generating function is the probability generating function of a Poisson process so

$$f(z) = e^{(1+s_{ben})(1-z)}.$$

Thus, we need to find the smallest non-negative solution, $z$, to

$$e^{(1+s_{ben})(1-z)} - z = 0.$$

We can use a second-order Taylor approximation to approximate the exponential so we have

$$1 + (1 + s_{ben})(z-1) + \frac{((1+s_{ben})(z-1))^2}{2} - z = 0.$$

which has the smallest non-negative solution of

$$z = \frac{s_{ben}^2 + 1}{(s_{ben} + 1)^2}.$$

If $s_{ben}$ is small, then we have

$$z \approx \frac{1}{1 + 2s_{ben}}.$$

which we can further approximate by doing

$$z \approx \frac{1}{1 + 2s_{ben}} \frac{1 - 2s_{ben}}{1 - 2s_{ben}} \approx 1 - 2s_{ben}. \tag{S16}$$

If $1 - 2s_{ben}$ is the extinction probability, then the fixation probability will be $2s_{ben}$.

## 2.2. Likelihood of reversion

We calculate the time for mutations to occur and fix in the population for demonstration in **Figure 3b**. For the sake of simplicity, we consider the weak-mutation strong selection regime (no clonal interference) in our theory but do include such dynamics in our simulations. Regardless, clonal interference will only minorly change the frequency of a revertant with high $s_{ben}$ in the population as multiple backgrounds will find the same reversion when $N_e$ is sufficiently high. The expected time to fixation given a mutation with selective coefficient $s_{ben}$ is estimated using

$$(1 + s_{ben})^t = N_e.$$

where $N_e$ is the census population size. The above implies

$$t\ln(1 + s_{ben}) = \ln(N_e).$$

And if $s_{ben}$ is small, then we have

$$t = \frac{\ln(N_e)}{s_{ben}}. \tag{S17}$$

Now we need to know the expected time for a fixing mutant to arise. First, we want the probability the mutation arises and fixes, which will be given by

$$1 - \left(1 - 2s_{ben}U_{ben}\right)^{N_e} \approx 2N_e s_{ben}U_{ben}.$$

This implies the expected time to arrive is approximately

$$\frac{1}{2N_e s_{ben}U_{ben}}. \tag{S18}$$

So, therefore, in the absence of clonal interference, the time for reversion to fix in the population depends more on the time for it to fix over time than the time for the reversion to arise (and is, therefore, less dependent on the mutation rate) if

$$\frac{1}{2N_e s_{ben}U_{ben}} < \frac{ln\left(N_e\right)}{s_{ben}}.$$

Alternatively written as

$$1 < 2N_e U_{ben} ln\left(N_e\right). \tag{S19}$$

Finally, note that the expected time to fixation is given by

$$\frac{1}{2N_e s_{ben}U_{ben}} + \frac{ln\left(N_e\right)}{s_{ben}}. \tag{S20}$$

The second term can be larger because of bottlenecks and clonal interference.

## 3.1 Reversion model with fluctuating loci

The reversion model can be derived in the following way. First, one can rewrite *Equation 2* as a generic negative feedback model in which mutations emerge and are purged from the population at a rate proportional to how many mutations have accumulated:

$$d\overline{N}_{transient} = R_{in}dt - P_{out}\overline{N}_{transient}dt. \tag{S21}$$

In this formulation, $R_{in}$ is simply the rate of accumulation of transient non-neutral nonsynonymous mutations per genome per unit time and $P_{out}$ is the loss rate of these mutations per unit time.

From this more general form, we can develop the reversion model. Similar to the original purifying selection model, it is possible to assume a variety of classes of mutations, but for simplicity, we only assume 1. If we make the following definitions for

$$R_{in} = \frac{n_{loci}}{\tau_{flip}}.$$

$$P_{out} = \frac{2}{\tau_{flip}}.$$

we can link *Equation S21* to a fluctuating loci model via

$$\begin{aligned} \frac{d\overline{N}_{transient}}{dt} &= R_{in} - P_{out}\overline{N}_{transient} = \frac{n_{loci}}{\tau_{flip}} - \frac{2}{\tau_{flip}}\overline{N}_{transient} \\ &= \frac{1}{\tau_{flip}}\left(n_{loci} - \overline{N}_{transient}\right) - \frac{1}{\tau_{flip}}\overline{N}_{transient} \end{aligned}. \tag{S22}$$

Here, $\tau_{flip}$ represents the average number of generations it takes for a *given* selective pressure on a locus to switch direction, and $n_{loci}$, the number of loci under fluctuating selective pressures. *Equation S22* can be more directly linked to a fluctuating loci model as we see the rate of nonsynonymous mutations is proportional to $n_{loci} - \overline{N}_{transient}$ (the number of loci unmutated) and the rate out is proportional to $\overline{N}_{transient}$ the number of loci mutated.

Solving similar to the purifying selection model, we have

$$\overline{N}_{transient}(t) = \frac{n_{loci}}{2}\left(1 - e^{-\frac{2t}{\tau_{flip}}}\right).$$

(S23)

This can then be used to obtain $d_N/d_S$

$$\frac{d_N}{d_S} = \frac{1}{3}\left(\frac{U_0}{U_S} + \frac{n_{loci}}{2U_S t}\left(1 - e^{-\frac{2t}{\tau_{flip}}}\right)\right).$$

Or equivalently

$$\frac{d_N}{d_S} = \frac{1}{3}\left(\frac{U_0}{U_S} + \frac{n_{loci}}{GdS}\left(1 - e^{-\frac{Gd_S}{U_S\tau_{flip}}}\right)\right).$$

If we set $\alpha = \frac{U_0}{3U_S}, \beta = \frac{G}{U_S\tau_{flip}}$, and $\gamma = \frac{n_{loci}}{3U_S\tau_{flip}}$ so that

$$\frac{d_N}{d_S} = \alpha + \frac{\gamma}{\beta dS}\left(1 - e^{-\beta d_S}\right).$$

(S24)

which is equivalent to *Equation S9* with one more free parameter.

## 3.2 Effect of compensatory mutations

We can further extend the theory to include compensatory mutations by calculating the expected value of $\overline{N}_{transient}$ as a Markov chain. First, let $v$ be the state vector for a locus under selection where the index of each row corresponds to the number of observed mutations currently at that locus and $A$ be the corresponding stochastic matrix with rows $i$ and columns $j$. First, we need to consider the probability the state does not change on a given generation (i.e., the diagonal of $A$). This will be $\frac{1}{\tau_{flip}}$. For every element on the diagonal of $A$, we thus have $A_{i=j} = 1 - \frac{1}{\tau_{flip}}$.

Supposing there is a state change, let us define $p$ as the probability there is an increase in observed mutations (a forward or compensatory mutation) and $1 - p$, the probability there is a decrease in observed mutations (a reversion). Then, $A_{j=i-1} = \frac{1}{\tau_{flip}}p$ and $A_{j=i+1} = \frac{1}{\tau_{flip}}(1 - p)$ (with the exception of $A_{i=1,j=0} = \frac{1}{\tau_{flip}}$). Finally, with $\Delta t$ being the number of time steps and assuming $v_0 = 1$ and $v_{i>0} = 0$, then

$$\overline{N}_{transient}(\Delta t) = n_{loci}\sum_{i\geq 0}\left(A^{\Delta t}v\right)_i \cdot i.$$

(S25)

If $p > 1 - p$, compensatory mutations are gained at a linear rate and the compensatory mutation rate will factor into the asymptotic $d_N/d_S$ value. Conversely, if $p < 1 - p$, the number of compensatory mutations is almost surely finite by the central limit theorem and hence will not factor into the asymptotic $d_N/d_S$. Finally, if $p = 1 - p$, this is the classic elementary random walk well known to deviate from the origin with $O(\sqrt{n})$. Interestingly, this is also sublinear and will not factor into asymptotic $d_N/d_S$.

## 3.3 Reversion model starting from equilibrium conditions

Our simulations and theory assume that the initial population starts with no forward mutations (i.e., WT) for simplicity. However, starting at equilibrium conditions does not impact the shape of the curve. The intuition here is that while starting from equilibrium enables the identification of reversions of initially transient mutations, these will be subsequently hidden by parallel evolution. Noting *Equation S23*, we have that for our simulations and base theory:

$$d_N = n_{loci}\left(1 - e^{-\frac{2t}{\tau_{flip}}}\right).$$

(S26)

We can show the same result occurs starting at equilibrium conditions. A Python script confirming the algebra is available on the GitHub repository. First, let there be three states a given locus can be in. The first state ($i = 1$) will be initial transient mutations, the second state ($i = 2$) will be ancestral alleles, and the third state ($i = 3$) will be subsequent transient mutations. We can then build a transition matrix:

$$A = \begin{pmatrix} -\dfrac{1}{\tau_{flip}} & 0 & 0 \\ \dfrac{1}{\tau_{flip}} & -\dfrac{1}{\tau_{flip}} & \dfrac{1}{\tau_{flip}} \\ 0 & \dfrac{1}{\tau_{flip}} & -\dfrac{1}{\tau_{flip}} \end{pmatrix}. \tag{S27}$$

We can then use the transition matrix to find the probability of being in a state at any given time via the matrix exponential, which is

$$e^{At} = \begin{pmatrix} e^{-\frac{t}{\tau_{flip}}} & 0 & 0 \\ \dfrac{1 - e^{-\frac{2t}{\tau_{flip}}}}{2} & \dfrac{1 + e^{-\frac{2t}{\tau_{flip}}}}{2} & \dfrac{1 - e^{-\frac{2t}{\tau_{flip}}}}{2} \\ \dfrac{1 + e^{-\frac{2t}{\tau_{flip}}}}{2} - e^{-\frac{t}{\tau_{flip}}} & \dfrac{1 - e^{-\frac{2t}{\tau_{flip}}}}{2} & \dfrac{1 + e^{-\frac{2t}{\tau_{flip}}}}{2} \end{pmatrix}. \tag{S28}$$

Now note that we can use $e^{At}(j, i)$ to find the probability of being in a given state $j$ after starting at initial state $i$. First, let us calculate expected $d_N$ at a locus that started from the ancestral allele and has subsequently diverged for time $t$. Take note that subsequent transient mutations are assumed to be distinct (i.e., they will always lead to at least one difference when compared to a different lineage).

$$d_N\left(inital\ ancestral\right) = e^{At}(2,2)\,e^{At}(3,2) + e^{At}(3,2)\,e^{At}(2,2) + 2e^{At}(3,2)\,e^{At}(3,2). \tag{S29}$$

Now let us calculate $d_N$ assuming the allele was initially a transient mutation.

$$+e^{At}(1,1)\,e^{At}(2,1) + e^{At}(2,1)\,e^{At}(1,1) + 2e^{At}(1,1)\,e^{At}(3,1) + 2e^{At}(3,1)\,e^{At}(1,1). \tag{S30}$$

Noting that we have assumed equal rates of forward and reverse adaptations throughout, the equilibrium would be composed of $n_{loci}/2$ initially ancestral loci and of $n_{loci}/2$ initially transient loci. Working through all of the algebra, we find

$$d_N = \frac{n_{loci}\left(d_N\left(inital\ ancestral\right) + d_N\left(inital\ transient\right)\right)}{2}.$$

$$d_N = \frac{n_{loci}\left(1 - e^{-\frac{2t}{\tau_{flip}}}\right) + n_{loci}\left(1 - e^{-\frac{2t}{\tau_{flip}}}\right)}{2}.$$

$$d_N = n_{loci}\left(1 - e^{-\frac{2t}{\tau_{flip}}}\right).$$

