## [Editor Report]

This valuable study addresses the interpretation of patterns of synonymous and nonsynonymous diversity in microbial genomes. The authors present solid theoretical and computational evidence that adaptive mutations that revert the amino acids to an earlier state can significantly impact the observed ratios of synonymous and nonsynonymous mutations in human commensal bacteria. This article will be of interest to microbiologists with a background in evolution and to researchers studying the human microbiome.

---

## [Decision Letter]

**Decision letter after peer review:**

Thank you for submitting your article "Reversions mask the contribution of adaptive evolution in microbiomes" for consideration by *eLife*. Your article has been reviewed by 3 peer reviewers, one of whom is a member of our Board of Reviewing Editors, and the evaluation has been overseen by Detlef Weigel as the Senior Editor.

Essential revisions (for the authors):

1) Address how recombination affects assumptions of your model, which invokes Muller's Ratchet to necessitate back–mutation. In effect, recombination of the wild–type allele is equivalent to back–mutation, but recombination can also revert or incorporate multiple linked variants at once and alter the assumptions of your model and analyses. As recombination tends to bring in many more mutations overall than occur in regions of a pair of genomes with asexual ancestry, the effects cannot be neglected. To what extent can this give rise to a similar dependence of dN/dS on dS as seen in the data?

2) Clarify the underlying assumptions of the model (e.g., the fitting parameters nloci and Tadapt), its theoretical results, and experimental applications. In addition, it clarifies the environmental conditions used for simulations, which could involve colonization of new hosts or environments, affect selection at many sites at once, and lead to clonal interference.

3) Evaluate the alternative that many compensatory mutations that could phenotypically revert an earlier mutation might overcome the exact specific reversion mutation in greater detail, including whether clonal interference between compensatory and reversion mutations would result in the mutations with the largest s – eg, as mentioned, reversion of a stop codon – being much more likely to sweep. Additional discussion of this important issue would be valuable.

*Reviewer #1 (Recommendations for the authors):*

This study makes a substantial contribution to our understanding of the molecular evolutionary dynamics of microbial genomes by proposing a model that incorporates relatively frequent adaptive reversion mutations. In many ways, this makes sense from my own experience with evolutionary genomic data of microbes, where reversions are surprisingly familiar as evidence of the immense power of selection in large populations.

One criticism is the reliance on one major data set of B. fragilis to test fits of these models, but this is relatively minor in my opinion and can be caveated by discussion of other relevant datasets for parallel investigation.

Another point is that this problem isn't as new as the manuscript indicates, see for example https://journals.asm.org/doi/10.1128/aem.02002–20.

Nonetheless, the paper succeeds by both developing theory and offering concrete parameters to illustrate the magnitudes of the problems that distinguish competing ideas, for example, the risk of mutational load posed in the absence of frequent back mutation.

Please expand the discussion of the novelty of incorporating (or overlooking) reversion mutations as well as the broader problems of dN/dS metrics for population–wide data where fixation is rare.

I'm otherwise enthusiastic about this study.

*Reviewer #2 (Recommendations for the authors):*

This manuscript asks how different forms of selection affect the patterns of genetic diversity in microbial populations. One popular metric used to infer signatures of selection is dN/dS, the ratio of nonsynonymous to synonymous distances between two genomes. Previous observations across many bacterial species have found dN/dS decreases with dS, which is a proxy for the divergence time. The most common interpretation of this pattern was proposed by Rocha et al. (2006), who suggested the excess in nonsynonymous mutations on short divergence times represent transient deleterious mutations that have not yet been purged by selection.

In this study, the authors propose an alternative model based on the population structure of human gut bacteria, in which dN is dominated by selective sweeps of SNPs that revert previous mutations within local populations. The authors argue that contrary to standard population genetics models, which are based on the population dynamics of large eukaryotes, the large populations in the human gut mean that reversions may be quite common and may have a large impact on evolutionary dynamics. They show that such a model can fit the decrease of dN/dS in time at least as well as the purifying selection model.

Strengths

The main strength of the manuscript is to show that adaptive sweeps in gut microbial populations can lead to small dN/dS. While previous work has shown that using dN/dS to infer the strength of selection within a population is problematic (see Kryazhimskiy and Plotkin, 2008, cited in the paper) the particular mechanism proposed by the authors is new to my knowledge. In addition, despite the known caveats, dN/dS values are still routinely reported in studies of microbial evolution, and so their interpretation should be of considerable interest to the community.

The authors provide compelling justification for the importance of adaptive reversions and make a good case that these need to be carefully considered by future studies of microbial evolution. The authors show that their model can fit the data as well as the standard model based on purifying selection and the parameters they infer appear to be plausible given known data. More generally, I found the discussion on the implications of traditional population genetics models in the context of human gut bacteria to be a valuable contribution of the paper.

Weaknesses

The authors argue that the purifying selection model would predict a gradual loss in fitness via Muller's ratchet. This is true if recombination is ignored, but this assumption is inconsistent with the data from Garud, et al. (2019) cited in the manuscript, who showed a significant linkage decrease in the bacteria also used in this study.

I also found that the data analysis part of the paper added little new to what was previously known. Most of the data comes directly from the Garud et al. study and the analysis is very similar as well. Even if other appropriate data may not currently be available, I feel that more could be done to test specific predictions of the model with more careful analysis.

Finally, I found the description of the underlying assumptions of the model and the theoretical results difficult to understand. I could not, for example, relate the fitting parameters nloci and Tadapt to the simulations after reading the main text and the supplement. In addition, it was not clear to me if simulations involved actual hosts or how the changes in selection coefficients for different sites was implemented. Note that these are not simply issues of exposition since the specific implementation of the model could conceivably lead to different results. For example, if the environmental change is due to the colonization of a different host, it would presumably affect the selection coefficients at many sites at once and lead to clonal interference. Related to this point, it was also not clear that the weak mutation strong selection assumption is consistent with the microscopic parameters of the model. The authors also mention that "superspreading" may somehow make a difference to the probability of maintaining the least loaded class in the purifying selection model, but what they mean by this was not adequately explained.

I see three main issues that would significantly improve the manuscript if addressed. The first is the issue of recombination, which undermines using Muller's ratchet to motivate the proposed model. Including the effects of recombination in the model would be a significant project in itself so I do not expect the authors to do that, but I do believe it is important that the issue be seriously addressed. In any case, I think the fact that the values of s implied by the purifying selection model are so small is already problematic, for many of the reasons discussed in the paper, so I would suggest focusing more on that. It might be more useful perhaps to demonstrate in simulations how even rare adaptive sweeps would affect the purging of deleterious mutations with small s, which is mentioned in the manuscript but not developed.

The second issue I found is the data analysis. It would be useful to check some of the hypotheses raised in the text directly from the data. For example, the authors mention that premature stop codons may preferentially be subject to reversions, which could be verified. Even if these analyses do not directly support the model, the authors can still comment on them and I believe it would significantly add to the paper.

Finally, I found the presentation of the model and simulations to be confusing and feel that it needs to be made significantly clearer for readers to be able to understand. In general, the supplement had considerable detail on standard results (Sec. 2.1 and 2.2 for example), but was very vague on the actual model itself (Sec. 3.1 and 3.2). If anything, this emphasis should be reversed. I've added some more specific comments below which might help with this issue. It should also be noted that the inferred values of s from any simple model will be effective parameters that may not be directly related to measurable parameters from experiments or observations. To give a very simple example, if selection coefficients are varying quickly in time, the relevant s for the accumulation of mutations would be the time average s, which may be considerably smaller than the typical value.

*Reviewer #3 (Recommendations for the authors):*

The diversity of bacterial species in the human gut microbiome is widely known, but the extensive diversity within each species is far less appreciated. Strains found in individuals on opposite sides of the globe can differ by as little as handfuls of mutations, while strains found in an individual's gut, or in the same household, might have a common ancestor tens of thousands of years ago. What are the evolutionary, ecological, and transmission dynamics that established and maintain this diversity?

The time, T, since the common ancestor of two strains, can be directly inferred by comparing their core genomes and finding the fraction of synonymous (non–amino acid changing) sites at which they differ: dS. With the per–site per–generation mutation rate, μ, and the mean generation times roughly known, this directly yields T (albeit with substantial uncertainty of the generation time.) A traditional way to probe the extent to which selection plays a role is to study pairs of strains and compare the fraction of non–synonymous (amino acid or stop–codon changing) sites, dN, at which the strains differ with their dS. Small dN/dS, as found between distantly related strains, is attributed to purifying selection against deleterious mutations dominating over mutations that have driven adaptive evolution. Large dN/dS as found in laboratory evolution experiments, is caused by beneficial mutations that quickly arise in large bacterial populations, and, with substantial selective advantages, per generation, can rise to high abundance fast enough that very few synonymous mutations arise in the lineages that take over the population.

A number of studies (including by Lieberman's group) have analyzed large numbers of strains of various dominant human gut species and studied how dN/dS varies. Although between closely related strains the variations are large – often much larger than attributable to just statistical variations – a systematic trend from dN/dS around unity or larger for close relatives to dN/dS ~ 0.1 for more distant relatives has been found in enough species that it is natural to conjecture a general explanation.

The conventional explanation is that, for close relatives, the effects of selection over the time since they diverged has not yet purged weakly deleterious mutations that arose by chance – roughly mutations with sT<1 – while since the common ancestor of more distantly related strains, there is plenty of time for most of those that arose to have been purged.

Torrillo and Lieberman have carried out an in–depth – sophisticated and quantitative – analysis of models of some of the evolutionary processes that shape the dependence of dN/dS on dS – and hence on their divergence time, T. They first review the purifying selection model and show that – even ignoring its inability to explain dN/dS > 1 for many closely related pairs – the model has major problems explaining the crossover from dN/dS somewhat less than unity to much smaller values as dS goes through – on a logarithmic scale – the 10^–4 range. The first problem, already seen in the infinite–population–size deterministic model, is that a very large fraction of non–synonymous mutations would have to have deleterious s's in the 10^–5 per generation range to fit the data (and a small fraction effectively neutral). As the s's are naturally expected (at least in the absence of quantitative analysis to the contrary) to be spread out over a wide range on a logarithmic scale of s, this seems implausible. But the authors go further and analyze the effects of fluctuations that occur even in the very large populations: ~ >10^12 bacteria per species in one gut, and 10^10 human guts globally. They show that Muller's ratchet – the gradual accumulation of weakly deleterious mutations that are not purged by selection – leads to a mutational meltdown with the parameters needed to fit the purifying selection model. In particular, with N_e the "effective population size" that roughly parametrizes the magnitude of stochastic birth–death and transition fluctuations, and U the total mutation rate to such deleterious mutations this occurs for U/s > log(sN_e) which they show would obtain with the fitted parameters.

Torrillo and Lieberman promise an alternate model: that there are a modest number of "loci" at which conditionally beneficial mutations can occur that are beneficial in some individual guts (or other environmental conditions) at some times, but deleterious in other (or the same) gut at other times. With the ancestors of a pair of strains having passed through one too many individuals and transmissions, it is possible for a beneficial mutation to occur and rise in the population, only later to be reverted by the beneficial inverse mutation. With tens of loci at which this can occur, they show that this process could explain the drop of dN/dS from short times – in which very few such mutations have occurred – to very long times by which most have flipped back and forth so that a random pair of strains will have the same nucleotide at such sites with 50% probability. Their qualitative analysis of a minimally simple model of this process shows that the bacterial populations are plenty big enough for such specific mutations to occur many times in each individual's gut, and with modest beneficials, to takeover. With a few of these conditionally beneficial mutations or reversions occurring during an individuals lifetime, they get a reasonably quantitative agreement with the dN/dS vs dS data with very few parameters. A key assumption of their model is that genetically exact reversion mutations are far more likely to takeover a gut population – and spread – than compensatory mutations which have a similar phenotypic–reversion effect: a mutation that is reverted does not show up in dN, while one that is compensated by another shows up as a two–mutation difference after the environment has changed twice.

Strengths:

The quantitative arguments made against the conventional purifying selection model are highly compelling, especially the consideration of multiple aspects that are usually ignored, including – crucially – how Muller's ratchet arises and depends on the realistic and needed–to–fit parameters; the effects of bottlenecks in transmission and the possibility that purifying selection mainly occurs then; and complications of the model of a single deleterious s, to include a distribution of selective disadvantages. Generally, the author's approach of focusing on the simplest models with as few as possible parameters (some roughly known), and then adding in various effects one–by–one, is outstanding and, in being used to analyze environmental microbial data, exceptional.

The reversion model the authors propose and study is a simple general one and they again explore carefully various aspects of it – including dynamics within and between hosts – and the consequent qualitative and quantitative effects. Again, the quantitive analysis of almost all aspects is exemplary. Although it is hard to make a compelling guess of the number of loci that are subject to alternating selection on the needed time–scales (years to centuries) they make a reasonable argument for a lower bound in terms of the number of known invertible promoters (that can genetically switch gene expression on and off).

Weaknesses:

The primary weakness of this paper is one that the author's are completely open about: the assumption that, collectively, any of possibly–many compensatory mutations that could phenotypically revert an earlier mutation, are less likely to arise and takeover local populations than the exact specific reversion mutation. While detailed analysis of this is, reasonably enough, beyond the scope of the present paper, more discussion of this issue would add substantially to this work. Quantitatively, the problem is that even a modest number of compensatory mutations occurring as the environmental pressures change could lead to enough accumulation of non–synonymous mutations that they could cause dN/dS to stay large – easily >1 – to much larger dS than is observed. If, say, the appropriate locus is a gene, the number of combinations of mutations that are better in each environment would play a role in how large dN would saturate to in the steady state (1/2 of n_loci in the author's model). It is possible that clonal interference between compensatory and reversion mutations would result in the mutations with the largest s – eg, as mentioned, reversion of a stop codon – being much more likely to take over, and this could limit the typical number of differences between quite well–diverged strains. However, the reversion and subsequent re–reversion would have to both beat out other possible compensatory mutations – naively less likely. I recommend that a few sentences in the Discussion be added on this important issue along with comments on the more general puzzle – at least to this reader! – as to why there appear to be so little adaptive genetic changes in core genomes on time scales of human lifetimes and civilization.

An important feature of gut bacterial evolution that is now being intensely studied is only mentioned in passing at the end of this paper: horizontal transfer and recombination of core genetic material. As this tends to bring in many more mutations overall than occur in regions of a pair of genomes with asexual ancestry, the effects cannot be neglected. To what extent can this give rise to a similar dependence of dN/dS on dS as seen in the data? Of course, such a picture begs the question as to what sets the low dN/dS of segments that are recombined –– often from genetic distances comparable to the diameter of the species.

The main substantive criticisms are summarized in "Weaknesses" section. Here are a few more minor points, and then presentational and pedagogical comments.

1) The analysis of the effects of a distribution of values of s is not convincing. A simple argument that for each s there is a crossover dS ~ mu/s, and approximating the corresponding portion of dN/dS vs dS as a step function at that crossover, gives a better approximation that using the harmonic mean: the rough distribution of s needed to fit, can then be eyeballed from the data – roughly by derivative d[dN/dS]/dS. (This point is roughly made – although hard to parse and confounded by uninformative details – in the Supplement).

2) For Muller's ratchet and deleterious loading: The time dependence computed from the deterministic analysis is rather misleading as it assumes – unreasonably – that the common ancestor had no deleterious mutations. The main point is the small fraction of the least–loaded subpopulation, W(t=infinity), being very small. The comparison of this with the appropriate N_e – whether N_e s exp(–U/s) is small or large – determining whether Muller's ratchet operates is the important point which needs emphasizing more: and this should be in the main text. [Note that for Muller's ratchet, using the harmonic mean s is roughly correct.] I would replace existing Eq(5) with an equation with this comparison in it (now just loosely in words around L 163) and refer to Ref(29) for analyses. Explain that if N is large enough or U small enough, reversions of the deleterious mutations will prevent Muller's ratchet.

Additionally, make clear that the model assumes additive effects of the deleterious mutations, and note that the average burden is small – and thus additive probably reasonable – even when large enough that Muller's ratchet is effective.

3) Some more discussion of "effective population size" is needed when introduced, in addition to that in the Discussion. This is especially true here, since – as noted in Discussion – it is often taken from the dS of a species via N_e = /mu: simply the overall T_MRCA in units of generations. The way the present paper uses N_e, is how it appears in modeling and thus controls several actual biological quantities: how much mutations can change abundance by drift, and how often beneficial mutations occur. This is thus a "real" effective population size. It is important to note, however, that with clonal interference (both in individuals and across human populations) different "N_e"'s are involved for different processes. A detailed discussion is not needed, but a couple of sentences (and a reference) in the main text is – around L165 – and a note in caption of Figure 2.

4) What is being assumed in various places (eg Figure 2) for the generation time is unclear.

5) The heuristic explanation for the time dependence of the reversion model is missing. T_adapt is rather confusing. A basic quantity – call it eg tau_change – is how often the conditions for a particular transient mutation change from beneficial to deleterious. This is the time that corresponds to the dS at which the crossover in the dN/dS data occurs – and indeed what is in the equations. T_adapt = tau_change/n_loci is the typical time between when any of these mutations switch. This determines dN. The distinction needs explaining.

6) The meaning of "loci" in counting the number of possible reverting mutations is unclear. Presumably what is meant is that at each locus there may be a number of roughly equivalent mutations – with each changing the phenotype so the others would no longer be beneficial – but to revert the phenotype needs to revert the specific one that occurred. This needs to be made clearer. Also, note that this would change the numbers somewhat as the beneficial mutation rate would be higher, but it has shown that analysis is not sensitive to that factor because of the large populations. Generally, a clearer statement that the beneficial s is the important quantity setting the time scales – log(Ns)/s – rather than the time for mutation to arise, is needed in the main text.

7) Effects of recombination: It would be good to add some more discussion on whether the effects of recombination alone (with an assumption about the effects of purifying selection on long time scales) could fit the data – as pointed out in "Weaknesses" section.

Is reversion by recombination likely to have a substantial role in dN/dS? I do not find the suggestion that multiple mutations could be simultaneously reverted by recombination at all believable. The further away the segment is, the more implausible is the scenario that compensatory mutations – let all other beneficial mutations with epistasis between them – have not occurred.

8) Generally, for each of the models and extensions used, make clear the number and set of parameters, and the assumptions. (Eg Figure 3)

Presentation:

Although the overall scenario, points made, and many of the explanations are quite clear, aspects of the presentation, figures, and captions are unclear or sloppy and detract from the paper's readability and potential impact. More specific and clear statements of the assumptions and model features are needed. And which of the mathematics to include or not in the main text, the supplement, or not at all, could be improved.

Notation:

– Mutation rates: mu_S etc are more often used per site: Here they are used for genome–wide instead of the more conventional – and better – U_S etc

– Use some notation for s of beneficial mutants – eg s_adapt.

– T_adapt see above

– N is used both for population size and numbers. With the use of n_loci, might be better to use n(t)'s – perhaps best with an overbar – for the average number of mutations that accumulate with time.

In many figures, notations such as "1.8e+01" or, in one, "quadrillion" are used: Use standard notation and, eg 10^1 mark instead of 1.8 times this.

Figure 1:

It is not clear what is meant by "adaptive genes" in the Zhao et al. data shown.

In the caption, say that crossover occurs at roughly dS ~ mu/s : this makes the role of s clear. In general, this kind of heuristic note is useful – independently of whether or not a reader is mathematically inclined!

L98: Make clear N_transient etc are averages. Spell out that for dS << 1, dS ~ 2 U_S T_MRCA, and this regime obtains for all the data.

L127: Spell out mu_S (or better U_S) in terms of the number of sites and generation time.

L150: the minus sign in the exponent is not visible (similarly in some other equations)

L187–194: It is unclear what is meant by "superspreaders" and what the 30,000X factor means. Is it that strains from one individual would have to spread to 30,000 others before reversions etc? The argument against superspreaders could be expanded beyond "seems unlikely".

L204–213: Equation (6) and surrounding text on general process, is not needed and confusing. Instead, the left side and last equality in Eq S23 should be pulled into the main text. Then can go directly to Eq(7) with an explanation that the n_loci/2 factor is simply the average number of differences between distantly related bacteria.

L259: Is the fitted T_adapt in generations? Also, give in years. And don't pretend to precision as "840", and later "110 years": give ranges.

Figure 4: (a): note in the caption that actual dN/dS > 1 and why.

Specify n_loci used.

(b) "discounting beneficial mutations" is unclear: are or aren't the reversions included in this statement?

L287: Explain what PAML is/assumes.

L321: Note that in the simulations with genetic classes, the dN, etc between two individuals cannot be directly kept track of. Correct?

L348: Meaning of "negative feedback" is unclear.

L353: As per main comments: more is needed on how tiny the rate of compensatory or other beneficial mutations would have to be to not raise α hugely.

L365: Explain "pseudogenized".

Methods and Theory:

Generally, much more than needed deriving standard results.

Wright–Fisher simulations: The notation here is very bad, with nearby letters being used for very different things – eg why not t instead of i which is in any case elsewhere used to mean something different? – and j_k used as it is but with j being an index. And generally, it is overly long and unclear.

L 673: what is RMSD?

L678: Explain PAML

dN/dS Theory: Much too much detail as most very standard Eg: certainly don't need all the derivation leading up to L742 equation.

L777: Make clear that α used in the main text is that with 0 subscript here.

L791–815 is confusing, not needed, and probably not valid. The point made in L817 is all that is needed – see comment (1) above.

L860: Say that sNW factor is mean fitness.

L895 is a classic result and not all derivation is needed.

L912: "has 2s probability of extinction" is unclear what it means: I presume something about the rate of ratchet. L915 equations, as noted in (2) should be in the main text.

Extinction and Fixation Probability:

This is all standard and not needed beyond quoting results.

L980: "Mutation is accessible if… " is very misleading. If L982 is satisfied, then there will be clonal interference between multiple mutations, and much of the analysis changes. But whether satisfied or not, the mutations can arise and fix: what changes is which will dominate the time to fixation as in L989

L1005: As noted, should pull part of this into the main text.

Implied Amount of Superspreading: extremely unclear. (and S26 is not even an equation)

SI Figures: There are probably more than needed and captions are often insufficient. Would be better to have a few paragraphs of text explaining them collectively, and the figures interspersed in the text.

Figure S4: What is "theory" curve?

Figure S5: Very unclear what was done or what modeled. Why is every 100,000 generations underestimated of bottleneck? What does "beneficial mutations are released" mean.

---

## [Author Response]

Essential revisions (for the authors):1) Address how recombination affects assumptions of your model, which invokes Muller's Ratchet to necessitate back–mutation. In effect, recombination of the wild–type allele is equivalent to back–mutation, but recombination can also revert or incorporate multiple linked variants at once and alter the assumptions of your model and analyses. As recombination tends to bring in many more mutations overall than occur in regions of a pair of genomes with asexual ancestry, the effects cannot be neglected. To what extent can this give rise to a similar dependence of dN/dS on dS as seen in the data?

We now discuss the effect of recombination on the purifying selection model on page 6 and in Figure 2—figure supplement 3. In short, we now show that reasonable levels of recombination cannot rescue the purifying selection model from Muller’s ratchet when *s* is so low and the influx of new deleterious mutations is so high. We thank the reviewers for prompting this improvement.

2) Clarify the underlying assumptions of the model (e.g., the fitting parameters nloci and Tadapt), its theoretical results, and experimental applications. In addition, it clarifies the environmental conditions used for simulations, which could involve colonization of new hosts or environments, affect selection at many sites at once, and lead to clonal interference.

More clarification has been given in the text and the methods regarding the reversion model simulations and fitting. We initially left too much of the implementation to the reader reviewing the Github code, and we apologize for this oversight.

3) Evaluate the alternative that many compensatory mutations that could phenotypically revert an earlier mutation might overcome the exact specific reversion mutation in greater detail, including whether clonal interference between compensatory and reversion mutations would result in the mutations with the largest s – eg, as mentioned, reversion of a stop codon – being much more likely to sweep. Additional discussion of this important issue would be valuable.

We have now added simulations that include the possibility of compensatory mutations, and we have updated both the text and added an additional Figure 4—figure supplement 3. In short, we show that adding compensatory mutations that do not completely restore original fitness (as would be expected following a loss-of-function mutation) leads to a strengthening of the reversion model and a lower required number of locally adaptive loci. We thank the reviewers for prompting this improvement.

Reviewer #1 (Recommendations for the authors):This study makes a substantial contribution to our understanding of the molecular evolutionary dynamics of microbial genomes by proposing a model that incorporates relatively frequent adaptive reversion mutations. In many ways, this makes sense from my own experience with evolutionary genomic data of microbes, where reversions are surprisingly familiar as evidence of the immense power of selection in large populations.One criticism is the reliance on one major data set of B. fragilis to test fits of these models, but this is relatively minor in my opinion and can be caveated by discussion of other relevant datasets for parallel investigation.

We analyze data from 10 species of the *Bacteroidales* family, and we compare it to a dataset of *Bacteroides fragilis.* We have now added a reference to a recent manuscript from our group showing phenotypic alteration by reversion of a stop codon and further breaking of the same pathway through stop codons in other genes in *Burkholderia dolosa* on page 7, and have added a new analysis of codon usage in support of the reversion model on page 9-10.

We have chosen not to analyze other species as there are no large data sets with rigorous and evenly-applied quality control across scales. We anticipate the reversion model would be able to fit the data in these cases. We now note that this work remains to be done in the discussion.

Another point is that this problem isn't as new as the manuscript indicates, see for example https://journals.asm.org/doi/10.1128/aem.02002–20.

Loo et al. puts forward an explanation similar to the purifying model proposed by Rocha et al, which we refute here. Quoting from Loo et al: “Our results confirm the observation that nonsynonymous SNPs are relatively elevated under shorter time periods and that purifying selection is more apparent over longer periods or during transmission.” While there is some linguistic similarity between the weak purifying model and our model of strong local adaptation model and strong adaptive reversion, we believe that the dynamical and predictive implications suggested by the reversion model are an important conceptual leap and correction to the literature. We now cite Loo et al. and additional works cited therein. We have updated the abstract, introduction, and discussion to further emphasize the distinction of the reversion model from previous models: namely the implication of the reversion model that long-time scale dN/dS hides dynamics.

Nonetheless, the paper succeeds by both developing theory and offering concrete parameters to illustrate the magnitudes of the problems that distinguish competing ideas, for example, the risk of mutational load posed in the absence of frequent back mutation.Please expand the discussion of the novelty of incorporating (or overlooking) reversion mutations as well as the broader problems of dN/dS metrics for population–wide data where fixation is rare.

We now better emphasize the distinction between the weak purifying selection model and reversion model in both the abstract, results, and discussion. We further emphasize that this work inverts the conclusion from prior works -- that dN/dS gives misleading interpretations of dynamics on long time scales, and that short-term measures (intra-population) of dN/dS may be more informative than long-term (inter-population) measures of dN/dS for dynamical inferences. Thank you for instigating this improvement.

I'm otherwise enthusiastic about this study.Reviewer #2 (Recommendations for the authors):This manuscript asks how different forms of selection affect the patterns of genetic diversity in microbial populations. One popular metric used to infer signatures of selection is dN/dS, the ratio of nonsynonymous to synonymous distances between two genomes. Previous observations across many bacterial species have found dN/dS decreases with dS, which is a proxy for the divergence time. The most common interpretation of this pattern was proposed by Rocha et al. (2006), who suggested the excess in nonsynonymous mutations on short divergence times represent transient deleterious mutations that have not yet been purged by selection.In this study, the authors propose an alternative model based on the population structure of human gut bacteria, in which dN is dominated by selective sweeps of SNPs that revert previous mutations within local populations. The authors argue that contrary to standard population genetics models, which are based on the population dynamics of large eukaryotes, the large populations in the human gut mean that reversions may be quite common and may have a large impact on evolutionary dynamics. They show that such a model can fit the decrease of dN/dS in time at least as well as the purifying selection model.StrengthsThe main strength of the manuscript is to show that adaptive sweeps in gut microbial populations can lead to small dN/dS. While previous work has shown that using dN/dS to infer the strength of selection within a population is problematic (see Kryazhimskiy and Plotkin, 2008, cited in the paper) the particular mechanism proposed by the authors is new to my knowledge. In addition, despite the known caveats, dN/dS values are still routinely reported in studies of microbial evolution, and so their interpretation should be of considerable interest to the community.The authors provide compelling justification for the importance of adaptive reversions and make a good case that these need to be carefully considered by future studies of microbial evolution. The authors show that their model can fit the data as well as the standard model based on purifying selection and the parameters they infer appear to be plausible given known data. More generally, I found the discussion on the implications of traditional population genetics models in the context of human gut bacteria to be a valuable contribution of the paper.

Thank you for the kind words and appreciation of the manuscript.

WeaknessesThe authors argue that the purifying selection model would predict a gradual loss in fitness via Muller's ratchet. This is true if recombination is ignored, but this assumption is inconsistent with the data from Garud, et al. (2019) cited in the manuscript, who showed a significant linkage decrease in the bacteria also used in this study.

We now investigate the effect of recombination on the purifying selection model on page 6 and in Figure 2—figure supplement 3. In short, we show that reasonable levels of recombination (obtained from literature r/m values) cannot rescue the purifying selection model from Muller’s ratchet when *s* is so low and the influx of new deleterious mutations is so high. We thank the reviewers for prompting this improvement.

I also found that the data analysis part of the paper added little new to what was previously known. Most of the data comes directly from the Garud et al. study and the analysis is very similar as well. Even if other appropriate data may not currently be available, I feel that more could be done to test specific predictions of the model with more careful analysis.

In addition to new analyses regarding recombination and compensatory mutations using the Garud et al. data set, we have now added two new analyses, both using *Bacteroides fragilis*. First, we show that de novo mutations in Zhao & Lieberman et al. dataset include an enrichment of premature stop codons (page 7). Second we show that genes expected to be under fluctuating selection in *B. fragilis* displays a significant closeness to stop codons, consistent with recent stop codons and reversions. We thank the reviewer for prompting the improvement.

Finally, I found the description of the underlying assumptions of the model and the theoretical results difficult to understand. I could not, for example, relate the fitting parameters nloci and Tadapt to the simulations after reading the main text and the supplement. In addition, it was not clear to me if simulations involved actual hosts or how the changes in selection coefficients for different sites was implemented. Note that these are not simply issues of exposition since the specific implementation of the model could conceivably lead to different results. For example, if the environmental change is due to the colonization of a different host, it would presumably affect the selection coefficients at many sites at once and lead to clonal interference. Related to this point, it was also not clear that the weak mutation strong selection assumption is consistent with the microscopic parameters of the model. The authors also mention that "superspreading" may somehow make a difference to the probability of maintaining the least loaded class in the purifying selection model, but what they mean by this was not adequately explained.

We apologize for leaving the specifics of the implementation from the paper and only accessible through the Github page and have corrected this. We have added a new section in the methods further detailing the reversion model and the specifics of how nloci and Tadapt (now tau_switch as of the edits) are implemented in the code.

The possibility for clonal interference is indeed included in the simulation. Switching is not correlated with transmissions in our main figure simulations (Figure 4a). When we run simulations in which transmission and selection are correlated, the results remain essentially the same, barring higher variance at lower divergences Figure 4-figure supplement 1. We have now clarified these points in the results, and have also better clarified the selection only at transmission model in the main results.

I see three main issues that would significantly improve the manuscript if addressed. The first is the issue of recombination, which undermines using Muller's ratchet to motivate the proposed model. Including the effects of recombination in the model would be a significant project in itself so I do not expect the authors to do that, but I do believe it is important that the issue be seriously addressed. In any case, I think the fact that the values of s implied by the purifying selection model are so small is already problematic, for many of the reasons discussed in the paper, so I would suggest focusing more on that. It might be more useful perhaps to demonstrate in simulations how even rare adaptive sweeps would affect the purging of deleterious mutations with small s, which is mentioned in the manuscript but not developed.

As mentioned above, we have updated the manuscript to include a section and SI figure detailing the effect of recombination on the purifying selection model.

We demonstrate the problem of rare adaptive sweeps on the purifying model in Figure 2—figure supplement 4. We have now expanded this paragraph to further emphasize this point.

The second issue I found is the data analysis. It would be useful to check some of the hypotheses raised in the text directly from the data. For example, the authors mention that premature stop codons may preferentially be subject to reversions, which could be verified. Even if these analyses do not directly support the model, the authors can still comment on them and I believe it would significantly add to the paper.

As mentioned above, we have added two new analyses regarding *B. fragilis,* frequency of stop codons, and codon usage which support the reversion model.

Finally, I found the presentation of the model and simulations to be confusing and feel that it needs to be made significantly clearer for readers to be able to understand. In general, the supplement had considerable detail on standard results (Sec. 2.1 and 2.2 for example), but was very vague on the actual model itself (Sec. 3.1 and 3.2). If anything, this emphasis should be reversed. I've added some more specific comments below which might help with this issue. It should also be noted that the inferred values of s from any simple model will be effective parameters that may not be directly related to measurable parameters from experiments or observations. To give a very simple example, if selection coefficients are varying quickly in time, the relevant s for the accumulation of mutations would be the time average s, which may be considerably smaller than the typical value.

As mentioned above, we now have fixed this oversight and added much more detail about the model.

Reviewer #3 (Recommendations for the authors):The diversity of bacterial species in the human gut microbiome is widely known, but the extensive diversity within each species is far less appreciated. Strains found in individuals on opposite sides of the globe can differ by as little as handfuls of mutations, while strains found in an individual's gut, or in the same household, might have a common ancestor tens of thousands of years ago. What are the evolutionary, ecological, and transmission dynamics that established and maintain this diversity?The time, T, since the common ancestor of two strains, can be directly inferred by comparing their core genomes and finding the fraction of synonymous (non–amino acid changing) sites at which they differ: dS. With the per–site per–generation mutation rate, μ, and the mean generation times roughly known, this directly yields T (albeit with substantial uncertainty of the generation time.) A traditional way to probe the extent to which selection plays a role is to study pairs of strains and compare the fraction of non–synonymous (amino acid or stop–codon changing) sites, dN, at which the strains differ with their dS. Small dN/dS, as found between distantly related strains, is attributed to purifying selection against deleterious mutations dominating over mutations that have driven adaptive evolution. Large dN/dS as found in laboratory evolution experiments, is caused by beneficial mutations that quickly arise in large bacterial populations, and, with substantial selective advantages, per generation, can rise to high abundance fast enough that very few synonymous mutations arise in the lineages that take over the population.A number of studies (including by Lieberman's group) have analyzed large numbers of strains of various dominant human gut species and studied how dN/dS varies. Although between closely related strains the variations are large – often much larger than attributable to just statistical variations – a systematic trend from dN/dS around unity or larger for close relatives to dN/dS ~ 0.1 for more distant relatives has been found in enough species that it is natural to conjecture a general explanation.The conventional explanation is that, for close relatives, the effects of selection over the time since they diverged has not yet purged weakly deleterious mutations that arose by chance – roughly mutations with sT<1 – while since the common ancestor of more distantly related strains, there is plenty of time for most of those that arose to have been purged.Torrillo and Lieberman have carried out an in–depth – sophisticated and quantitative – analysis of models of some of the evolutionary processes that shape the dependence of dN/dS on dS – and hence on their divergence time, T. They first review the purifying selection model and show that – even ignoring its inability to explain dN/dS > 1 for many closely related pairs – the model has major problems explaining the crossover from dN/dS somewhat less than unity to much smaller values as dS goes through – on a logarithmic scale – the 10^–4 range. The first problem, already seen in the infinite–population–size deterministic model, is that a very large fraction of non–synonymous mutations would have to have deleterious s's in the 10^–5 per generation range to fit the data (and a small fraction effectively neutral). As the s's are naturally expected (at least in the absence of quantitative analysis to the contrary) to be spread out over a wide range on a logarithmic scale of s, this seems implausible. But the authors go further and analyze the effects of fluctuations that occur even in the very large populations: ~ >10^12 bacteria per species in one gut, and 10^10 human guts globally. They show that Muller's ratchet – the gradual accumulation of weakly deleterious mutations that are not purged by selection – leads to a mutational meltdown with the parameters needed to fit the purifying selection model. In particular, with N_e the "effective population size" that roughly parametrizes the magnitude of stochastic birth–death and transition fluctuations, and U the total mutation rate to such deleterious mutations this occurs for U/s > log(sN_e) which they show would obtain with the fitted parameters.Torrillo and Lieberman promise an alternate model: that there are a modest number of "loci" at which conditionally beneficial mutations can occur that are beneficial in some individual guts (or other environmental conditions) at some times, but deleterious in other (or the same) gut at other times. With the ancestors of a pair of strains having passed through one too many individuals and transmissions, it is possible for a beneficial mutation to occur and rise in the population, only later to be reverted by the beneficial inverse mutation. With tens of loci at which this can occur, they show that this process could explain the drop of dN/dS from short times – in which very few such mutations have occurred – to very long times by which most have flipped back and forth so that a random pair of strains will have the same nucleotide at such sites with 50% probability. Their qualitative analysis of a minimally simple model of this process shows that the bacterial populations are plenty big enough for such specific mutations to occur many times in each individual's gut, and with modest beneficials, to takeover. With a few of these conditionally beneficial mutations or reversions occurring during an individuals lifetime, they get a reasonably quantitative agreement with the dN/dS vs dS data with very few parameters. A key assumption of their model is that genetically exact reversion mutations are far more likely to takeover a gut population – and spread – than compensatory mutations which have a similar phenotypic–reversion effect: a mutation that is reverted does not show up in dN, while one that is compensated by another shows up as a two–mutation difference after the environment has changed twice.Strengths:The quantitative arguments made against the conventional purifying selection model are highly compelling, especially the consideration of multiple aspects that are usually ignored, including – crucially – how Muller's ratchet arises and depends on the realistic and needed–to–fit parameters; the effects of bottlenecks in transmission and the possibility that purifying selection mainly occurs then; and complications of the model of a single deleterious s, to include a distribution of selective disadvantages. Generally, the author's approach of focusing on the simplest models with as few as possible parameters (some roughly known), and then adding in various effects one–by–one, is outstanding and, in being used to analyze environmental microbial data, exceptional.The reversion model the authors propose and study is a simple general one and they again explore carefully various aspects of it – including dynamics within and between hosts – and the consequent qualitative and quantitative effects. Again, the quantitive analysis of almost all aspects is exemplary. Although it is hard to make a compelling guess of the number of loci that are subject to alternating selection on the needed time–scales (years to centuries) they make a reasonable argument for a lower bound in terms of the number of known invertible promoters (that can genetically switch gene expression on and off).

We are very grateful for the reviewer’s kind words and careful reading.

Weaknesses:The primary weakness of this paper is one that the author's are completely open about: the assumption that, collectively, any of possibly–many compensatory mutations that could phenotypically revert an earlier mutation, are less likely to arise and takeover local populations than the exact specific reversion mutation. While detailed analysis of this is, reasonably enough, beyond the scope of the present paper, more discussion of this issue would add substantially to this work. Quantitatively, the problem is that even a modest number of compensatory mutations occurring as the environmental pressures change could lead to enough accumulation of non–synonymous mutations that they could cause dN/dS to stay large – easily >1 – to much larger dS than is observed. If, say, the appropriate locus is a gene, the number of combinations of mutations that are better in each environment would play a role in how large dN would saturate to in the steady state (1/2 of n_loci in the author's model). It is possible that clonal interference between compensatory and reversion mutations would result in the mutations with the largest s – eg, as mentioned, reversion of a stop codon – being much more likely to take over, and this could limit the typical number of differences between quite well–diverged strains. However, the reversion and subsequent re–reversion would have to both beat out other possible compensatory mutations – naively less likely. I recommend that a few sentences in the Discussion be added on this important issue along with comments on the more general puzzle – at least to this reader! – as to why there appear to be so little adaptive genetic changes in core genomes on time scales of human lifetimes and civilization.

We now directly consider compensatory mutations (page 9, SI text 3.2, and Figure 4-figure supplement 3). We show that as long as true reversions are more likely than compensatory mutations overall, (adaptive) nonsynonymous mutations will still tend to revert towards their initial state and not contribute to asymptotic dN/dS, and show that true reversions are expected in a large swath of parameter space. Thank you for motivating this improvement!

We note in the discussion that directional selection could be incorporated into the parameter α (assuming even more of the genome is deleterious) on page 11.

An important feature of gut bacterial evolution that is now being intensely studied is only mentioned in passing at the end of this paper: horizontal transfer and recombination of core genetic material. As this tends to bring in many more mutations overall than occur in regions of a pair of genomes with asexual ancestry, the effects cannot be neglected. To what extent can this give rise to a similar dependence of dN/dS on dS as seen in the data? Of course, such a picture begs the question as to what sets the low dN/dS of segments that are recombined –– often from genetic distances comparable to the diameter of the species.

We now discuss the effect of recombination on the purifying selection model on page 6 and in Figure 2—figure supplement 3. In short, we now show that reasonable levels of recombination cannot rescue the purifying selection model from Muller’s ratchet when *s* is so low and the influx of new deleterious mutations is so high. We thank the reviewers for prompting this improvement

The main substantive criticisms are summarized in "Weaknesses" section. Here are a few more minor points, and then presentational and pedagogical comments.1) The analysis of the effects of a distribution of values of s is not convincing. A simple argument that for each s there is a crossover dS ~ mu/s, and approximating the corresponding portion of dN/dS vs dS as a step function at that crossover, gives a better approximation that using the harmonic mean: the rough distribution of s needed to fit, can then be eyeballed from the data – roughly by derivative d[dN/dS]/dS. (This point is roughly made – although hard to parse and confounded by uninformative details – in the Supplement).

We agree that an argument involving a step function is more straightforward. We have changed the analysis to now use this logic for explaining the effects of a distribution of s and have streamlined this section more generally. Thank you for the improvement.

2) For Muller's ratchet and deleterious loading: The time dependence computed from the deterministic analysis is rather misleading as it assumes – unreasonably – that the common ancestor had no deleterious mutations. The main point is the small fraction of the least–loaded subpopulation, W(t=infinity), being very small. The comparison of this with the appropriate N_e – whether N_e s exp(–U/s) is small or large – determining whether Muller's ratchet operates is the important point which needs emphasizing more: and this should be in the main text. [Note that for Muller's ratchet, using the harmonic mean s is roughly correct.] I would replace existing Eq(5) with an equation with this comparison in it (now just loosely in words around L 163) and refer to Ref(29) for analyses. Explain that if N is large enough or U small enough, reversions of the deleterious mutations will prevent Muller's ratchet.Additionally, make clear that the model assumes additive effects of the deleterious mutations, and note that the average burden is small – and thus additive probably reasonable – even when large enough that Muller's ratchet is effective.

We replaced equation 5 with S17 which puts into context the size needed for Muller’s Ratchet to be prevented and suggested ref(29) (now ref (40)) for further analysis. We also noted the assumption of additive mutation effects. Thank you for the improvement.

3) Some more discussion of "effective population size" is needed when introduced, in addition to that in the Discussion. This is especially true here, since – as noted in Discussion – it is often taken from the dS of a species via N_e = /mu: simply the overall T_MRCA in units of generations. The way the present paper uses N_e, is how it appears in modeling and thus controls several actual biological quantities: how much mutations can change abundance by drift, and how often beneficial mutations occur. This is thus a "real" effective population size. It is important to note, however, that with clonal interference (both in individuals and across human populations) different "N_e"'s are involved for different processes. A detailed discussion is not needed, but a couple of sentences (and a reference) in the main text is – around L165 – and a note in caption of Figure 2.

We now more thoroughly discuss effective population size in the main text when introduced (page 4-5) and refer to it in the caption of Figure 2. Thank you for the improvement.

4) What is being assumed in various places (eg Figure 2) for the generation time is unclear.

We assume about one generation a day. We added this in the text and apologize for the confusion.

5) The heuristic explanation for the time dependence of the reversion model is missing. T_adapt is rather confusing. A basic quantity – call it eg tau_change – is how often the conditions for a particular transient mutation change from beneficial to deleterious. This is the time that corresponds to the dS at which the crossover in the dN/dS data occurs – and indeed what is in the equations. T_adapt = tau_change/n_loci is the typical time between when any of these mutations switch. This determines dN. The distinction needs explaining.

We agree that using tau_change (which we call Tau_flip) over T_adapt provides a more intuitive understanding of the time dependence and have changed text and equations accordingly. We thank the reviewer for prompting this improvement.

6) The meaning of "loci" in counting the number of possible reverting mutations is unclear. Presumably what is meant is that at each locus there may be a number of roughly equivalent mutations – with each changing the phenotype so the others would no longer be beneficial – but to revert the phenotype needs to revert the specific one that occurred. This needs to be made clearer. Also, note that this would change the numbers somewhat as the beneficial mutation rate would be higher, but it has shown that analysis is not sensitive to that factor because of the large populations. Generally, a clearer statement that the beneficial s is the important quantity setting the time scales – log(Ns)/s – rather than the time for mutation to arise, is needed in the main text.

Yes, this is the correct interpretation. We have clarified this in the text and figure legend. In the simulations, the beneficial forward mutation rate is higher than the reversion rate to take this into account, and this has been made clear in the new methods section detailing how the reversion mutations function.

We have also adjusted the text and Figure 3b to emphasize the importance of s vs. mutation rate in setting time to reversion. Thanks for prompting the improvement.

7) Effects of recombination: It would be good to add some more discussion on whether the effects of recombination alone (with an assumption about the effects of purifying selection on long time scales) could fit the data – as pointed out in "Weaknesses" section.

We now discuss the effect of recombination on the purifying selection model on page 8 and in Figure 2—figure supplement 3. In short, we now show that reasonable levels of recombination cannot rescue the purifying selection model from Muller’s ratchet when *s* is so low and the influx of new deleterious mutations is so high.

Is reversion by recombination likely to have a substantial role in dN/dS? I do not find the suggestion that multiple mutations could be simultaneously reverted by recombination at all believable. The further away the segment is, the more implausible is the scenario that compensatory mutations – let all other beneficial mutations with epistasis between them – have not occurred.

We have now clarified this part of the discussion. Recombination would be helpful in the reversion model if multiple mutations had occurred in the same gene over longer periods of time, and the gene being completely replaced with the ancestral version would be beneficial upon environmental change. This is particularly likely if epistasis enables only 2+ step mutants to be beneficial upon environmental change. We hope the revised version is more convincing.

8) Generally, for each of the models and extensions used, make clear the number and set of parameters, and the assumptions. (Eg Figure 3)

We apologize for the confusion and have added more detail to the figure captions.

Presentation:Although the overall scenario, points made, and many of the explanations are quite clear, aspects of the presentation, figures, and captions are unclear or sloppy and detract from the paper's readability and potential impact. More specific and clear statements of the assumptions and model features are needed. And which of the mathematics to include or not in the main text, the supplement, or not at all, could be improved.Notation:– Mutation rates: mu_S etc are more often used per site: Here they are used for genome–wide instead of the more conventional – and better – U_S etc

Done.

– Use some notation for s of beneficial mutants – eg s_adapt.

Done.

– T_adapt see above

Done.

– N is used both for population size and numbers. With the use of n_loci, might be better to use n(t)'s – perhaps best with an overbar – for the average number of mutations that accumulate with time.

We added in an overbar (thank you for the suggestion!), but we continue to use capitalize N for nonsynonymous mutations for the sake of matching to dN.

In many figures, notations such as "1.8e+01" or, in one, "quadrillion" are used: Use standard notation and, eg 10^1 mark instead of 1.8 times this.

Done.

Figure 1:It is not clear what is meant by "adaptive genes" in the Zhao et al. data shown.

Clarification has been added. We apologize for the confusion.

In the caption, say that crossover occurs at roughly dS ~ mu/s : this makes the role of s clear. In general, this kind of heuristic note is useful – independently of whether or not a reader is mathematically inclined!

We have made the changes. Thank you for the suggestion.

L98: Make clear N_transient etc are averages. Spell out that for dS << 1, dS ~ 2 U_S T_MRCA, and this regime obtains for all the data.

We have made the changes. Thank you for the suggestion.

L127: Spell out mu_S (or better U_S) in terms of the number of sites and generation time.

For fitting the data, since dS=#S / number of S sites, only the mutation rate per base pair matters. We now further discuss the molecular clock on page 3 to provide better clarity. U_N does depend on the number of sites and we have now made this more explicit on page 4.

L150: the minus sign in the exponent is not visible (similarly in some other equations)

We are unsure how to remedy this as they do appear on our version of the document.

L187–194: It is unclear what is meant by "superspreaders" and what the 30,000X factor means. Is it that strains from one individual would have to spread to 30,000 others before reversions etc? The argument against superspreaders could be expanded beyond "seems unlikely".

We apologize for the confusion and have added more clarification. We now only discuss superspreaders more directly in the Results section relating to the purifying selection model for clarity.

L204–213: Equation (6) and surrounding text on general process, is not needed and confusing. Instead, the left side and last equality in Eq S23 should be pulled into the main text. Then can go directly to Eq(7) with an explanation that the n_loci/2 factor is simply the average number of differences between distantly related bacteria.

We agree and have made the changes accordingly and left some of the notes on the more general process to the SI. Thank you for the improvement.

L259: Is the fitted T_adapt in generations? Also, give in years. And don't pretend to precision as "840", and later "110 years": give ranges.

We have clarified and given ranges. Thank you for the suggestion.

Figure 4: (a): note in the caption that actual dN/dS > 1 and why.

Done.

Specify n_loci used.

Done.

(b) "discounting beneficial mutations" is unclear: are or aren't the reversions included in this statement?

Reworded and clarified.

L287: Explain what PAML is/assumes.

Done.

L321: Note that in the simulations with genetic classes, the dN, etc between two individuals cannot be directly kept track of. Correct?

Correct, this is now noted.

L348: Meaning of "negative feedback" is unclear.

Reworded.

L353: As per main comments: more is needed on how tiny the rate of compensatory or other beneficial mutations would have to be to not raise α hugely.

See new section on compensatory mutations.

L365: Explain "pseudogenized".

Changed word to nonfunctional.

Methods and Theory:Generally, much more than needed deriving standard results.Wright–Fisher simulations: The notation here is very bad, with nearby letters being used for very different things – eg why not t instead of i which is in any case elsewhere used to mean something different? – and j_k used as it is but with j being an index. And generally, it is overly long and unclear.

We have reworked the indices to make it more clear. Thank you for the suggestion.

L 673: what is RMSD?

Root mean square deviation. This has been added in the text.

L678: Explain PAML

Clarification has been added.

dN/dS Theory: Much too much detail as most very standard Eg: certainly don't need all the derivation leading up to L742 equation.

Yes, the theory is purposely very highly detailed. The intent is to make it available to a wide audience of readers including those who may not know mathematics beyond basic calculus, as the majority of the theory doesn’t use any very advanced techniques. We hope that this helps interest more readers in doing theory. We now clarify this intent.

L777: Make clear that α used in the main text is that with 0 subscript here.

Done.

L791–815 is confusing, not needed, and probably not valid. The point made in L817 is all that is needed – see comment (1) above.

We do agree it is probably not needed and now make the point in a quicker more straightforward way that depends upon the step function approximation as suggested.

L860: Say that sNW factor is mean fitness.

Done.

L895 is a classic result and not all derivation is needed.

Yes, this was included in an attempt to make the work self contained for readers without a background in population genetics. We hope that this is useful to readers. We now add the word classic result to make sure the historical context of this is understood.

L912: "has 2s probability of extinction" is unclear what it means: I presume something about the rate of ratchet. L915 equations, as noted in (2) should be in the main text.

Yes, clarified. The equation has been moved.

Extinction and Fixation Probability:This is all standard and not needed beyond quoting results.

Yes, this was included in an attempt to make the work self contained for readers without a background in population genetics.

L980: "Mutation is accessible if… " is very misleading. If L982 is satisfied, then there will be clonal interference between multiple mutations, and much of the analysis changes. But whether satisfied or not, the mutations can arise and fix: what changes is which will dominate the time to fixation as in L989

Changed language used, thank you.

L1005: As noted, should pull part of this into the main text.

Done.

Implied Amount of Superspreading: extremely unclear. (and S26 is not even an equation)

We decided to remove this part of the SI as it was not adding much and clarify more in the main text.

SI Figures: There are probably more than needed and captions are often insufficient. Would be better to have a few paragraphs of text explaining them collectively, and the figures interspersed in the text.

Most of the SI figures are more for checks of robustness than major points about theory and are more meant to be looked at in context of the main text. Thus, we kept them as are though we expanded upon the captions to add more detail.

Figure S4: What is "theory" curve?

Changed label to infinite population in top panel. Removed theory line in the second panel upon review as it is confusing since the asymptote occurs because of mutation accumulation which is not modeled by theory.

Figure S5: Very unclear what was done or what modeled. Why is every 100,000 generations underestimated of bottleneck? What does "beneficial mutations are released" mean.

We apologize for the confusion. This is just running the simulation for the reversion model, which assumes a population transmitting through bottlenecks and adaptations under the assumptions of the purifying selection model (ie. so remove the reversions part, have less frequent forward mutations, and make deleterious mutations have a smaller deleterious effect). We have added more to the figure captions to explain.